# TMEM16 and OSCA/TMEM63 proteins share a conserved potential to permeate ions and phospholipids

**Augustus J Lowry[1†], Pengfei Liang[1†], Mo Song[2], Yuichun Wan[1], Zhen-Ming Pei[3], Huanghe Yang[1,4]\*, Yang Zhang[1,2]\*†**

[1]Department of Biochemistry, Duke University School of Medicine, Durham, United States; [2]Institute of Molecular Physiology, Shenzhen Bay Laboratory, Guangdong, China; [3]Department of Biology, Duke University, Durham, United States; [4]Department of Neurobiology, Duke University School of Medicine, Durham, United States

**\*For correspondence:**
huanghe.yang@duke.edu (HY);
zhangyang@szbl.ac.cn (YZ)

†These authors contributed equally to this work

**Competing interest:** The authors declare that no competing interests exist.

## eLife Assessment

This **important** study advances our understanding of the mechanisms controlling lipid flux and ion permeation in the TMEM16 and OSCA/TMEM63 family channels. The study provides **compelling** new evidence indicating that side chains along the TM4/6 interface play a key role in gating lipid and ion fluxes in these channels. The authors suggest that the transmembrane channel/scramblase family proteins may have originally functioned as scramblases but lost this capacity over evolution.

**Abstract** The calcium-activated TMEM16 proteins and the mechanosensitive/osmolarity-activated OSCA/TMEM63 proteins belong to the Transmembrane Channel/Scramblase (TCS) super-family. Within the superfamily, OSCA/TMEM63 proteins, as well as TMEM16A and TMEM16B, are thought to function solely as ion channels. However, most TMEM16 members, including TMEM16F, maintain an additional function as scramblases, rapidly exchanging phospholipids between leaflets of the membrane. Although recent studies have advanced our understanding of TCS structure–function relationships, the molecular determinants of TCS ion and lipid permeation remain unclear. Here, we show that single mutations along the transmembrane helix (TM) 4/6 interface allow non-scrambling TCS members to permeate phospholipids. In particular, this study highlights the key role of TM 4 in controlling TCS ion and lipid permeation and offers novel insights into the evolution of the TCS superfamily, suggesting that, like TMEM16s, the OSCA/TMEM63 family maintains a conserved potential to permeate ions and phospholipids.

## Introduction

Ion channels and phospholipid scramblases catalyze the passive flux of ions and phospholipids down their respective chemical gradients. Compared to ion channels, both our understanding of scramblases and the evolutionary origins of phospholipid scrambling are underdeveloped. Discoveries of two scramblase families—the TMEM16 calcium-activated phospholipid scramblases (CaPLSases) and XKR caspase-dependent phospholipid scramblases—have revealed roles for scramblases in processes such as blood coagulation (*Suzuki et al., 2010*; *Yang et al., 2012*; *Fujii et al., 2015*), angiogenesis (*Shan et al., 2024*), cell death signaling (*Kunzelmann et al., 2019*; *Suzuki et al., 2013a*), phagocytosis (*Suzuki et al., 2013a*), cell migration (*Jacobsen et al., 2013*; *Chang et al., 2017*), membrane repair (*Griffin et al., 2016*; *Chandra et al., 2019*), microparticle release (*Fujii et al., 2015*), cell–cell fusion (*Whitlock et al., 2018*; *Zhang et al., 2020*; *Zhang et al., 2022*), and viral infection (*Zaitseva*

*et al., 2017*; *Nanbo et al., 2018*). Among identified scramblases, TMEM16 CaPLSases are the most extensively studied at the molecular level (*Le et al., 2021*). The TMEM16 family (*Figure 1—figure supplement 1*) was originally classified as calcium-activated chloride channels (CaCCs) based on the first-discovered members, TMEM16A and TMEM16B (*Caputo et al., 2008*; *Yang et al., 2008*; *Schroeder et al., 2008*). Most remaining members, however, exhibit CaPLSase activity, with TMEM16F representing the canonical CaPLSase (*Suzuki et al., 2013b*; *Figure 1a*). Uniquely, CaPLSases also function as non-selective ion channels along with their non-specific permeability to phospholipids (*Yang et al., 2012*; *Grubb et al., 2013*; *Wong et al., 2013*; *Yu et al., 2015*; *Di Zanni et al., 2018*; *Le et al., 2019b*). Substrate permeation is facilitated by calcium-induced conformational changes causing the clamshell-like separation of TMs 4 and 6 to facilitate ion and phospholipid permeation (*Le et al., 2019a*; *Lam et al., 2021*; *Lam and Dutzler, 2021*; *Figure 1—figure supplement 1b–d*). These conformational changes further catalyzed phospholipid permeation by thinning the membrane near the permeation pathway (*Falzone et al., 2019*; *Falzone et al., 2022*).

The TMEM16 family together with the OSCA/TMEM63 family and TMC family (*Keresztes et al., 2003*; *Hahn et al., 2009*; *Medrano-Soto et al., 2018*; *Ballesteros et al., 2018*) collectively form the Transmembrane Channel/Scramblase (TCS) superfamily (*Le et al., 2021*). Despite the conserved 10-TM architectural core (*Jojoa-Cruz et al., 2018*; *Liu et al., 2018*; *Zhang et al., 2018*; *Maity et al., 2019*; *Jeong et al., 2022*), phospholipid permeability has not been experimentally demonstrated in mechanosensitive OSCA/TMEM63 or TMC proteins, raising the intriguing question of how TCS proteins discriminate between ion and ion/phospholipid substrates. Interestingly, studies have shown that the TMEM16A CaCC can be genetically modified to enable phospholipid permeability, either by substituting TMEM16F domains as small as 15 amino acids (*Yu et al., 2015*), or through single mutations at the hydrophobic gate (*Le et al., 2019a*). These findings suggest a modest energetic barrier for scramblase activity and lead us to hypothesize that evolutionary relatives of the TMEM16 family may maintain a conserved potential for both ion and phospholipid permeation.

To test this hypothesis, we introduced single mutations along TM 4 of TMEM16F, TMEM16A, OSCA1.2, and TMEM63A at sites near the corresponding hydrophobic gate of TMEM16F (*Le et al., 2019a*). Mutations along the TM 4/6 interface in TMEM16F and TMEM16A resulted in constitutive phospholipid scramblase activity. Strikingly, equivalent TM 4 mutations in OSCA1.2 and TMEM63A also converted these ion channels into phospholipid scramblases, which were either constitutive or activated by osmotic stimulation. Individual mutations also resulted in gain-of-function (GOF) ion channel activity, suggesting that these mutations may disrupt a conserved activation gate in these evolutionarily related proteins. Together, our findings (1) advance the mechanistic understanding of gating and substrate permeation in the TMEM16 and OSCA/TMEM63 families, (2) underscore a key design principle for phospholipid scramblases, and (3) support our hypothesis that the TCS superfamily maintains a conserved potential for both ion and phospholipid permeation.

## Results

### TM 4 mutations result in a constitutively active TMEM16F scramblase

Our previous study identified the activation gate of TMEM16F, consisting of hydrophobic residues from TMs 4, 5, and 6, which collectively govern ion and phospholipid permeation (*Le et al., 2019a*). Mutations to the activation gate result in GOF CaPLSases, some of which are constitutively activated without requiring calcium (*Le et al., 2019a*). The hydrophobic gate represents the most constricted point along the TM 4/6 interface in TMEM16F and mutations along this interface likely alter substrate permeation and gating. Structural and computational studies on the fungal *Nh*TMEM16 (*Kalienkova et al., 2019*) and *Af*TMEM16 (*Falzone et al., 2019*) orthologs, as well as human TMEM16K (*Bushell et al., 2019*) further support a clamshell-like gating model (*Le et al., 2019a*) whereby scrambling activity is promoted by separation of the TM 4/6 interface (*Le et al., 2021*; *Figure 1—figure supplement 1b–d*). The conformational transition involves the N-terminal half of TM 4, which bends away from TM 6, and the C-terminal half of TM 6, which collapses onto the calcium-binding sites formed together with TMs 7 and 8 (*Le et al., 2021*). Given that the N-termini of TM 4 are largely hydrophobic among TMEM16 CaPLSases (*Figure 1—figure supplement 1f*) and mutations at F518 lead to constitutive TMEM16F activity (*Le et al., 2019a*), we hypothesized that introducing a charged side chain along the TM 4/6 interface might disrupt the hydrophobic gate and result in TMEM16

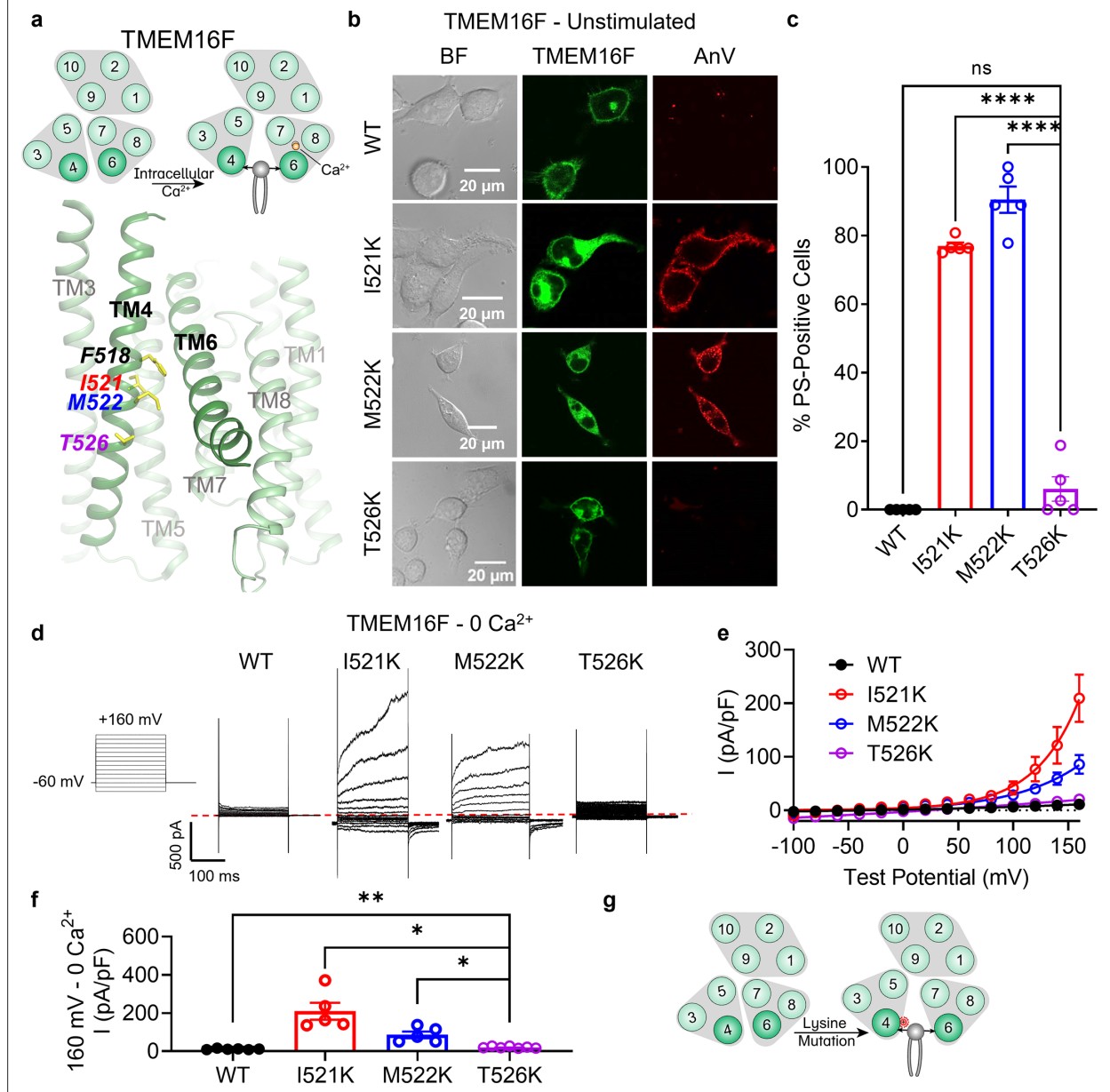

**Figure 1.** Lysine mutations along TM 4 enable TMEM16F channel and scramblase activities in the absence of calcium stimulation. (**a**) Top: TMEM16F is a calcium-activated phospholipid scramblase. Bottom: TM 4 mutant locations mapped on TMEM16F CaPLSase structure with side chains shown as yellow sticks (PDB 6QPB). (**b**) Representative images of TMEM16F knockout (KO) HEK293T cells expressing eGFP-tagged TMEM16F wildtype (WT), I521K, M522K, and T526K (center column). CF 594-conjugated Annexin V (AnV, right column) labeled phosphatidylserine (PS) exposing cells. BF denotes bright field images (left column). (**c**) Quantification of the percentage of cells with AnV labeling for TMEM16F WT ($n = 5$), I521K ($n = 5$), M522K ($n = 5$), and T526K ($n = 5$) transfected cells (*Figure 1—source data 1*). Values were derived from images of biological replicates. Statistical comparisons to T526K were done using unpaired $t$-tests with Welch's correction (ns: $p > 0.05$, ****$p < 0.0001$). (**d**) Representative current recordings and (**e**) current–voltage (*I–V*) relationships (*Figure 1—source data 1*) of whole-cell patches from TMEM16F KO HEK293T cells expressing eGFP-tagged TMEM16F WT ($n = 6$), I521K ($n = 5$), M522K ($n = 5$), and T526K ($n = 7$). Currents were elicited by the voltage protocol shown with the pipette solution containing 5 mM EGTA(Ethylene glycol tetraacetic acid). Dotted line denotes zero current. (**f**) Quantification of current at +160 mV (*Figure 1—source data 1*). Currents in (**e**) and (**f**) were normalized to cell capacitance. Statistical comparisons to T526K were done using unpaired $t$-tests with Welch's correction (*$p < 0.05$, **$p < 0.01$). (**g**) Lysine mutations along TM 4 in TMEM16F enable spontaneous phospholipid permeation in the absence of calcium. All error bars represent standard error of the mean (SEM) calculated from replicate images or independent patches.

The online version of this article includes the following source data and figure supplement(s) for figure 1:

**Source data 1.** Results from the lysine mutations on TM 4 of TMEM16F.

**Figure supplement 1.** Conformational changes in transmembrane helices (TMs) 4 and 6 are associated with TMEM16 scramblase activities.

*Figure 1 continued on next page*

*Figure 1 continued*

**Figure supplement 2.** Alternative side chains at I521 cause differential phosphatidylserine (PS) exposure.

**Figure supplement 2—source data 1.** Results from I521A and I521E of TMEM16F.

scramblases with GOF scramblase activity. To test this hypothesis, we overexpressed eGFP-tagged mutant constructs in TMEM16F knockout (KO) HEK293T cells and used an established scramblase assay (*Le et al., 2019b*; *Le et al., 2019a*; *Le et al., 2020*) that detects phosphatidylserine (PS) exposure using fluorophore-tagged Annexin V (AnV) as a reporter (*Figure 1b*). In the absence of calcium stimulation, PS is predominantly in the inner leaflet of the plasma membrane of TMEM16F wildtype (WT) expressing cells and is therefore not labeled by extracellular AnV. Similar to TMEM16F F518K (*Le et al., 2019a*), overexpressing the single lysine mutations TMEM16F I521K and M522K led to spontaneous, global exposure of PS on the plasma membrane without requiring calcium stimulation (*Figure 1b, c*). In contrast, T526K, which is one helical turn below I521 and M522 (*Figure 1a*), caused minimal spontaneous PS exposure (*Figure 1b, c*). We further tested whether I521K, M522K, and T526K have GOF ion channel activity using whole-cell patch clamp. In the absence of calcium, I521K and M522K, but not WT TMEM16F or T526K, showed robust depolarization-activated outward rectifying currents (*Figure 1d–f*). Our imaging and patch clamp experiments thus demonstrate that I521K and M522K are functionally expressed on the plasma membrane and their GOF phospholipid scramblase activity results in spontaneous, global PS exposure on the cell surface. Conversely, despite apparent plasma membrane localization, T526K exhibited strongly attenuated GOF ion channel and scramblase activities. The TM 4/6 interaction is notably less prominent near T526 (*Figure 1a*), two helical turns below the hydrophobic gate residue, F518 (*Le et al., 2019b*; *Figure 1a*) and may represent a lower limit for this charge-induced effect. On the other hand, I521 and M522 are approximately one helical turn below the hydrophobic gate (*Figure 1a*) supporting the idea that charged mutations along the TM 4/6 interface promote gate opening and substrate permeation (*Figure 1g*).

Our previous study also showed that glutamate, but not alanine, substitution at F518 led to spontaneous scramblase activity in TMEM16F (*Le et al., 2019a*). To assess whether alternative side chains cause spontaneous scrambling below the hydrophobic gate in TMEM16F, we tested I521A and I521E. I521E led to spontaneous, global PS exposure in all transfected cells, whereas I521A failed to cause spontaneous PS exposure (*Figure 1—figure supplement 2*). This result is consistent with our previous finding, providing additional support that charged TM 4 mutations disrupt the TM 4/6 interface.

## I611K on TM 6 results in a constitutively active TMEM16F scramblase

We next tested whether a TM 6 mutation along the TM 4/6 interface could promote scrambling. Directly adjacent to I521 and M522 are a pair of glycine residues on TM 6 that are proposed to function as a hinge during calcium-dependent activation (*Paulino et al., 2017*). We therefore chose to introduce a lysine mutation one helical turn above this site at I611, a residue adjacent to the pore-facing inner gate residue I612 (*Le et al., 2019b*; *Figure 2a*). Comparable to I521K (*Figure 1b, c*), I521E (*Figure 1—figure supplement 2*), and M522K (*Figure 1b, c*), I611K resulted in spontaneous, global PS exposure (*Figure 2b, c*). Likewise, I611K-expressing cells exhibited robust, depolarization-activated outward rectifying currents in the absence of calcium (*Figure 2d–f*). Together with our previous report that I612K is a GOF phospholipid scramblase and ion channel (*Le et al., 2019a*), I611K further supports that charged mutations in TM 6 near the activation gate can also promote substrate permeation in TMEM16F (*Figure 2g*).

## TM 4 lysine mutations convert TMEM16A into a constitutively active scramblase

TMEM16A is a CaCC (*Caputo et al., 2008*; *Yang et al., 2008*; *Schroeder et al., 2008*) without scramblase activity (*Yu et al., 2015*; *Le et al., 2019a*; *Figure 3a–c*). We previously reported that a single TM 4 lysine mutation, L543K, at the hydrophobic gate of TMEM16A (*Figure 3a*) allows the CaCC to permeate phospholipids spontaneously, analogous to the F518K mutation in TMEM16F (*Le et al., 2019a*). Thus, we reasoned that lysine mutations equivalent to TMEM16F I521K and M522K might also result in spontaneous phospholipid permeability. To test this hypothesis, we overexpressed eGFP-tagged TMEM16A I546K and I547K mutants in TMEM16F KO HEK293T cells and assessed

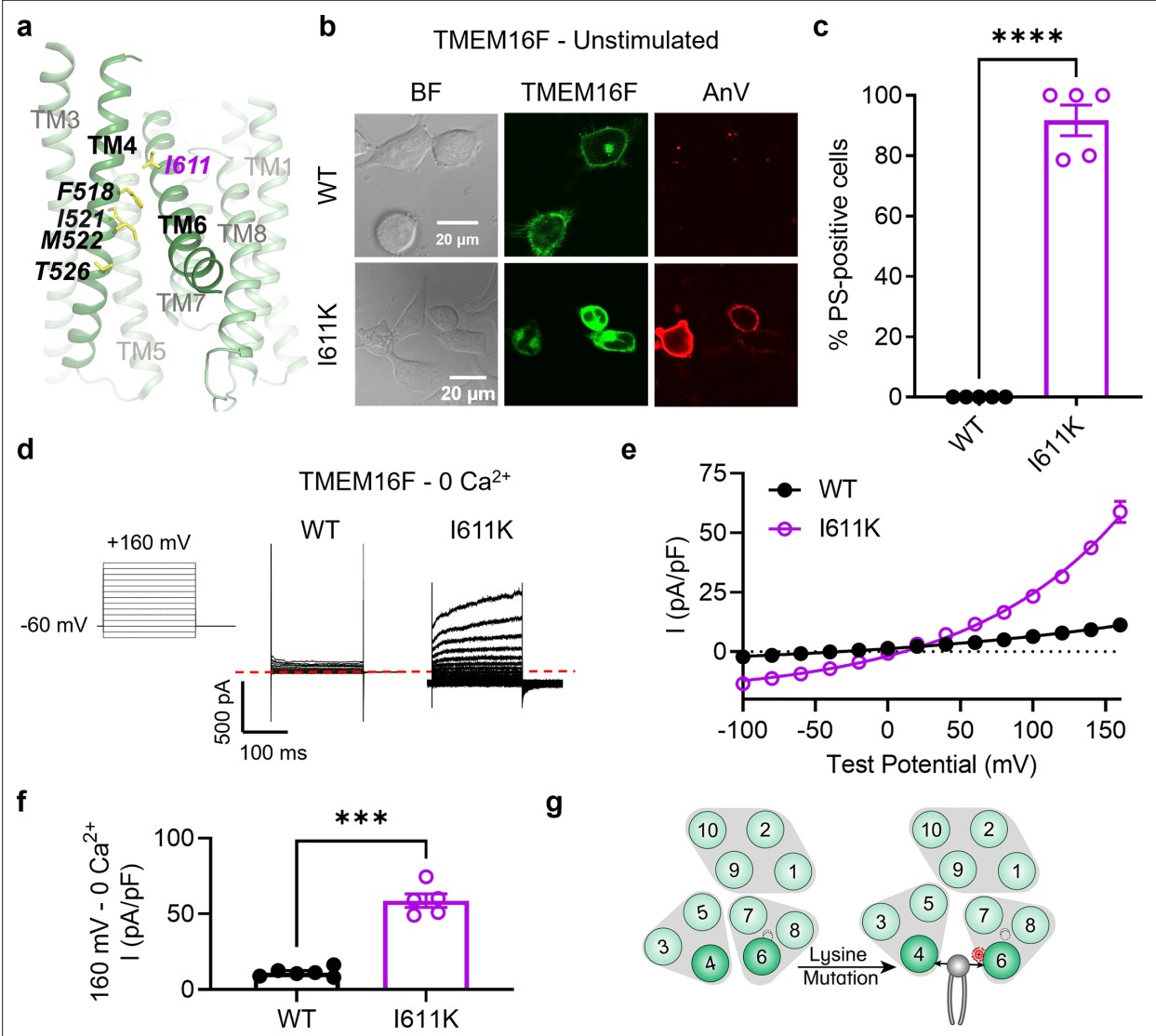

**Figure 2.** I611K on TM 6 enables TMEM16F channel and scramblase activities in the absence of calcium stimulation. (**a**) I611 highlighted on the TMEM16F CaPLSase structure with side chains shown as yellow sticks (PDB 6QPB). (**b**) Representative images of TMEM16F knockout (KO) HEK293T cells expressing eGFP-tagged TMEM16F wildtype (WT) and I611K (center column). CF 594-conjugated Annexin V (AnV, right column) labeled phosphatidylserine (PS) exposing cells. BF denotes bright field images (left column). (**c**) Quantification of the percentage of cells with AnV labeling for TMEM16F WT (*n* = 5) and I611K (*n* = 5) transfected cells (*Figure 2—source data 1*). Values were derived from images of biological replicates. Statistical comparison was done using an unpaired *t*-test with Welch's correction (****p < 0.0001). (**d**) Representative current recordings and (**e**) current–voltage (*I–V*) relationships (*Figure 2—source data 1*) of whole-cell patches from TMEM16F KO HEK293T cells expressing eGFP-tagged TMEM16F WT (*n* = 6) and I611K (*n* = 5). Currents were elicited by the voltage protocol shown with the pipette solution containing 5 mM EGTA. Dotted line denotes zero current. (**f**) Quantification of current at +160 mV (*Figure 2—source data 1*). Currents in (**e**) and (**f**) were normalized to cell capacitance. Statistical comparison was done using an unpaired *t*-test with Welch's correction (***p < 0.001). (**g**) A lysine mutation on TM 6 in TMEM16F enables spontaneous phospholipid permeation in the absence of calcium. All error bars represent standard error of the mean (SEM) calculated from replicate images or independent patches.

The online version of this article includes the following source data for figure 2:

**Source data 1.** Results from I611K on TM 6 of TMEM16F.

their ability to expose PS on the plasma membrane with confocal microscopy. Similar to TMEM16A L543K (*Le et al., 2019a*), I546K- and I547K-expressing cells exhibit spontaneous, global PS exposure (*Figure 3b, c*), analogous to the equivalent TMEM16F mutations I521K and M522K (*Figure 1—figure supplement 1f*; *Figure 1b, c*). Similarly, whole-cell patch clamp revealed GOF ion channel activity at depolarizing potentials, even in the absence of calcium stimulation (*Figure 3d–f*). In contrast, E551K,

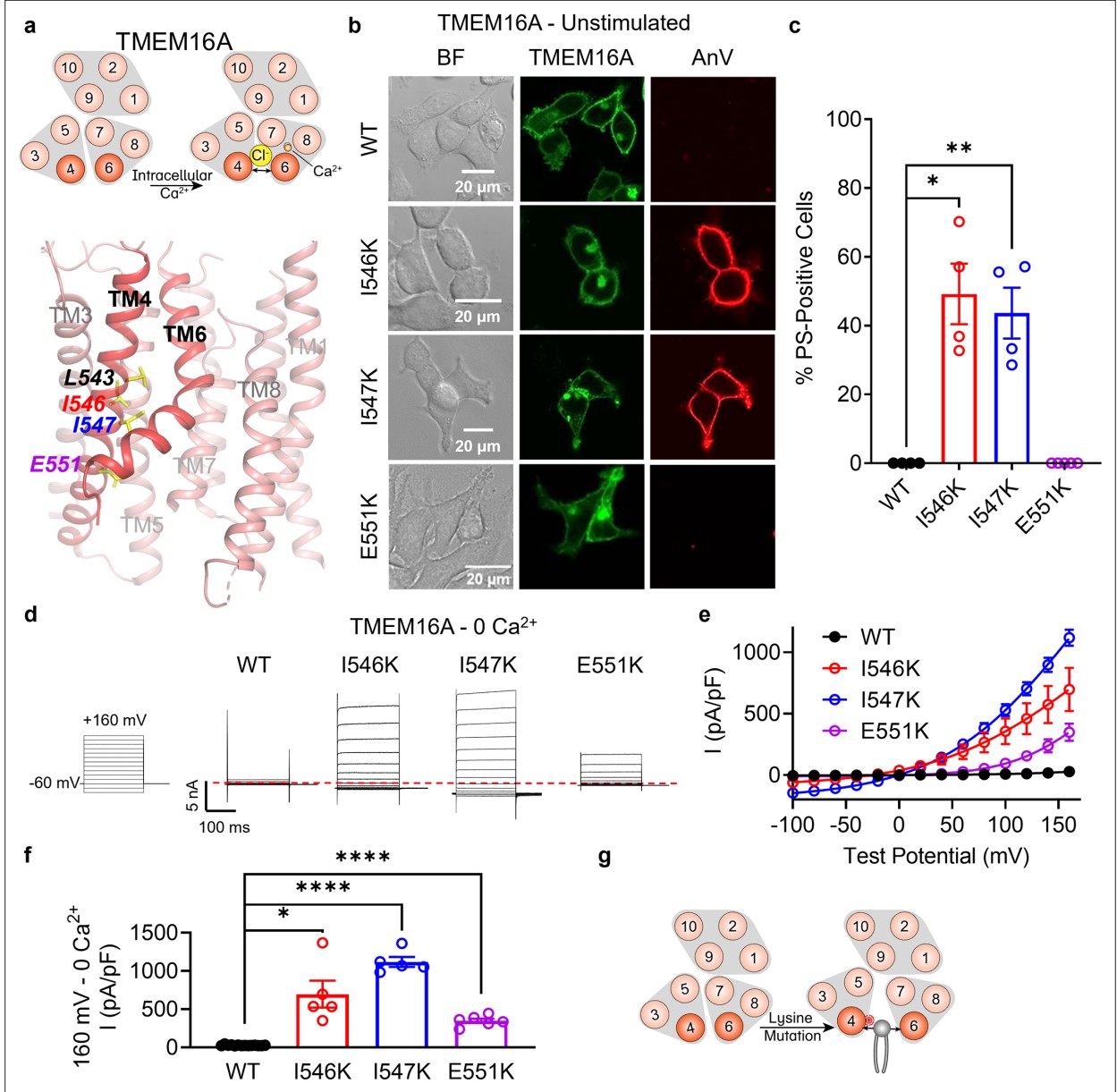

**Figure 3.** Lysine mutations along TM 4 enable TMEM16A channel and scramblase activities in the absence of calcium stimulation. (**a**) Top: TMEM16A is a calcium-activated chloride channel (CaCC). Bottom: TM 4 mutant locations mapped on TMEM16A CaCC structure with side chains shown as yellow sticks (PDB 5OYG). (**b**) Representative images of TMEM16F knockout (KO) HEK293T cells expressing eGFP-tagged TMEM16A wildtype (WT), I546K, I547K, and E551K (center column). CF 594-conjugated Annexin V (AnV, right column) labeled phosphatidylserine (PS) exposing cells. BF denotes bright field images (left column). (**c**) Quantification of the percentage of cells with AnV labeling for TMEM16A WT ($n = 4$), I546K ($n = 4$), I547K ($n = 4$), and E551K ($n = 5$) transfected cells (**Figure 3—source data 1**). Values were derived from images of biological replicates. Statistical comparisons were done using unpaired *t*-tests with Welch's correction (*$p < 0.05$, **$p < 0.01$). (**d**) Representative whole-cell current recordings and (**e**) current–voltage (*I–V*) relationships (**Figure 3—source data 1**) of whole-cell patches from TMEM16F KO HEK293T cells expressing eGFP-tagged TMEM16A WT ($n = 14$), I546K ($n = 5$), I547K ($n = 5$), and E551K ($n = 6$). Currents were elicited by the voltage protocol shown with the pipette containing 5 mM EGTA. Dotted line denotes zero current. (**f**) Quantification of current at +160 mV (**Figure 3—source data 1**). Currents in (**e**) and (**f**) were normalized to cell capacitance with the mean ± SEM calculated from independent patches. Statistical comparisons were done using unpaired *t*-tests with Welch's correction (*$p < 0.05$, ****$p < 0.0001$). (**g**) Lysine mutations along TM 4 in TMEM16A enable spontaneous phospholipid permeation in the absence of calcium. All error bars represent standard error of the mean (SEM) calculated from replicate images or independent patches.

The online version of this article includes the following source data and figure supplement(s) for figure 3:

**Source data 1.** Results from the lysine mutations on TM 4 of TMEM16A.

**Figure supplement 1.** Ani9 attenuates ion channel and phospholipid scramblase activities of TMEM16A I546K.

**Figure supplement 1—source data 1.** Results from Ani9 effects on TMEM16A I546K.

equivalent to T526K in TMEM16F, failed to exhibit PS exposure, but showed modest ion channel activity at depolarizing potentials in the absence of calcium, perhaps reflecting the gate is open sufficiently for ion but not phospholipid permeation. In total, these results suggest that TM 4 mutations in this region destabilize the interface to endow the TMEM16A CaCC with GOF ion channel activity, and, in the case of I546K and I547K, constitutive phospholipid permeability (*Figure 3g*).

## Ani9, a selective TMEM16A inhibitor, attenuates I546K ion and phospholipid permeation

We then tested whether Ani9, a selective, extracellular TMEM16A inhibitor (*Seo et al., 2016*), could attenuate phospholipid and ion permeation through I546K. Similar to our previous observation of TMEM16A L543K (*Le et al., 2019a*), chronic incubation of Ani9 at 10 μM also prevented spontaneous PS exposure for I546K (*Figure 3—figure supplement 1d*). Interestingly, 10 μM Ani9 reversibly inhibited voltage-elicited TMEM16A I546K currents by over 50% (*Figure 3—figure supplement 1a–c*). These results demonstrate that although the TM 4 GOF mutations alter TMEM16A substrate permeation, they retain their sensitivity to Ani9 inhibition (*Figure 3—figure supplement 1e*). This further supports that the TM 4 GOF mutations of TMEM16A mediate the spontaneous, global PS exposure.

## L438K on TM 4 converts OSCA1.2 into a constitutively active scramblase

We next applied our approach to a TMEM16 relative from the OSCA/TMEM63 family (*Figure 1—figure supplement 1a*), which was first discovered in plants as a family of mechanosensitive and osmolarity-activated cation non-selective ion channels (*Yuan et al., 2014*; *Figure 4a*). We hypothesized that analogous lysine mutations on TM 4 along the TM 4/6 interface of OSCA/TMEM63 proteins would result in GOF channels that may also permeate phospholipids. Within the family, we chose OSCA1.2 from *Arabidopsis thaliana* due to previous structural (*Jojoa-Cruz et al., 2018*) and biophysical (*Murthy et al., 2018*) characterization demonstrating that the channel is activated directly by membrane tension. Overexpressing eGFP-tagged OSCA1.2 WT in TMEM16F KO HEK293T cells did not induce PS exposure, demonstrating that OSCA1.2 WT lacks spontaneous phospholipid scramblase activity (*Figure 4b*). We then introduced TM 4 lysine mutations at analogous sites, as identified by structural (*Figure 4a*) and sequence alignment (*Figure 1—figure supplement 1f*). Strikingly, a single-point mutation, L438K, causes cells overexpressing the OSCA1.2 mutant to exhibit spontaneous and global PS exposure (*Figure 4b, c*). This mirrors our results with both TMEM16F (*Figure 1*) and TMEM16A (*Figure 3*), suggesting that the L438K mutation allows the OSCA1.2 channel to permeate phospholipids. Next, we used inside-out patch clamp to examine if L438K also enhances OSCA1.2 channel activity. We found that the mutant significantly left-shifts the conductance–voltage (G–V) relationship (*Figure 4e, f*) and accelerates channel activation kinetics (*Figure 4g*) compared to WT. Under −50 mmHg of pressure, L438K has a half-maximal voltage ($V_{0.5}$) of 66.7 ± 3.7 mV, while the WT $V_{0.5}$ is nearly 108.7 ± 5.6 mV (*Figure 4f*), underscoring that this mutation also promotes channel gating. Together, these experiments show that, like TMEM16A, a single lysine mutation near the putative gate allows the mechanosensitive and osmolarity-activated OSCA1.2 channel to become spontaneously permeable to phospholipids (*Figure 4h*).

To help visualize how L438K alters the TM 4/6 interface of OSCA1.2, we created a homology model of the mutant using the Swiss-PDB server and inserted it into a phospholipid membrane with the CHARMM-GUI webserver (*Jo et al., 2008*). After equilibration, we ran 900 ns atomistic molecular dynamics (MD) simulations for both WT and L438K in GROMACS, using a combination of conventional and metadynamics simulations. Remarkably, the L438K trajectory showed considerably increased hydration of the pore region compared to WT (*Figure 4—figure supplement 1b*). This enhanced hydration is attributed to a modest expansion of the TM 4/6 interface throughout the simulation (*Figure 4—figure supplement 1a*). Together with our functional results, these MD simulations reinforce our hypothesis that disruption of the TM 4/6 interface of OSCA1.2 promotes GOF ion and phospholipid permeation.

One helical turn above OSCA1.2 L438, at the corresponding site to the activation gate residue F518 in TMEM16F, is an endogenous lysine at position 435 (*Figure 4—figure supplement 2a*). This positioning places two positively charged residues in close proximity within TM 4 of the OSCA1.2 L438K mutant. Previous studies on model transmembrane helical peptides have demonstrated that

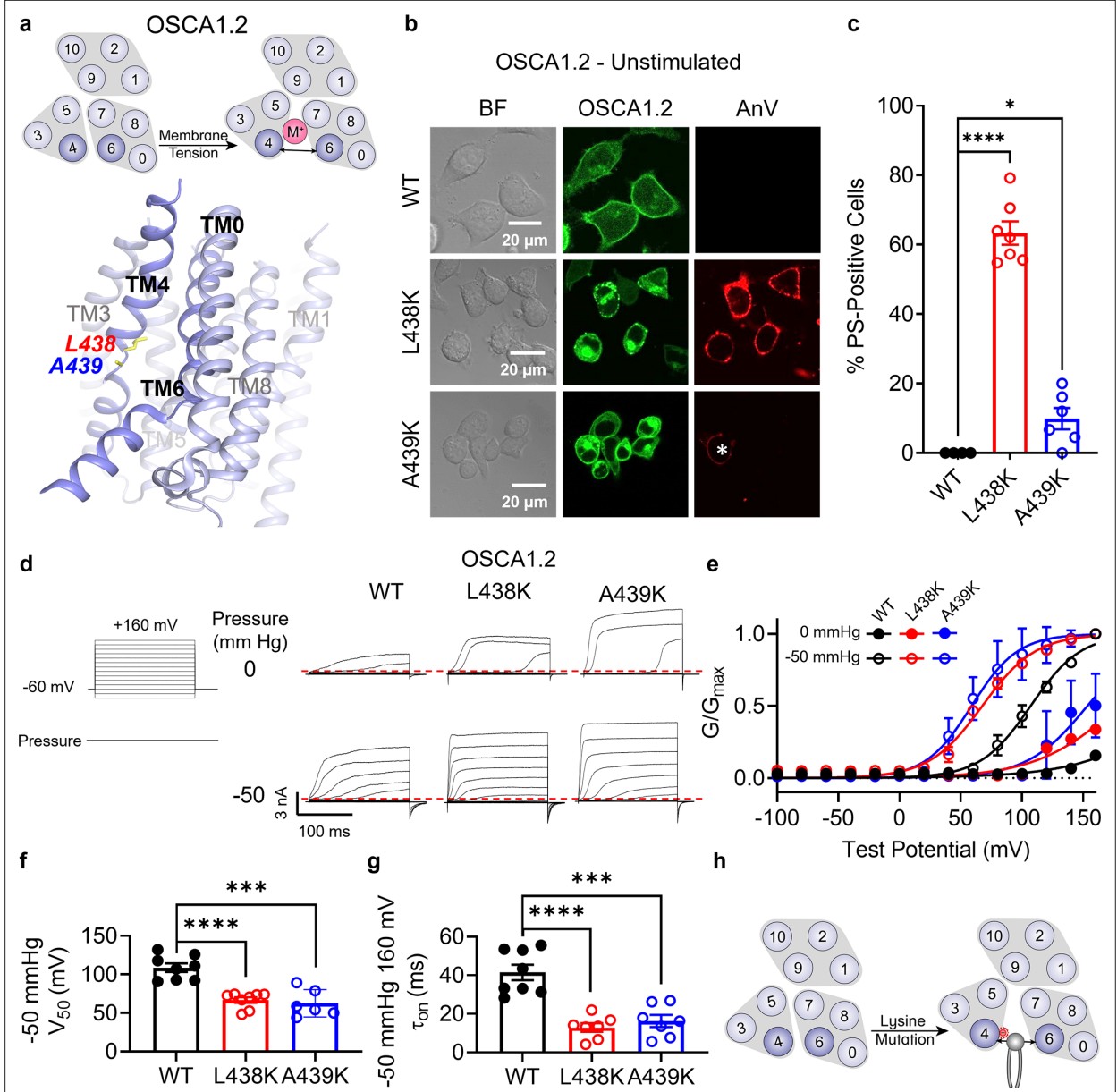

**Figure 4.** Lysine mutations along TM 4 enable OSCA1.2 channel and scramblase activities. (**a**) Top: OSCA1.2 is a cation non-selective ion channel gated by membrane tension. Bottom: TM 4 mutant locations mapped onto the TM 4/6 interface of OSCA1.2 (PDB 6MGV) with key residues shown as yellow sticks. (**b**) Representative images of TMEM16F knockout (KO) HEK293T cells expressing eGFP-tagged OSCA1.2 wildtype (WT), L438K, or A439K mutants (center column). CF 594-conjugated Annexin V (AnV) (right column) labeled phosphatidylserine (PS) exposing cells. BF denotes bright field images (left column). Asterisk highlights a PS-positive cell for the A439K mutant. (**c**) Quantification of the percentage of cells with AnV labeling for OSCA1.2 WT ($n$ = 4), L438K ($n$ = 7), and A439K-transfected cells ($n$ = 6) (**Figure 4—source data 1**). Statistical comparisons were conducted with unpaired $t$-tests with Welch's correction (*p < 0.05, ****p < 0.0001). (**d**) Representative current recordings and (**e**) normalized conductance–voltage ($G$–$V$) relationships of inside-out patches from TMEM16F KO HEK293T cells expressing eGFP-tagged OSCA1.2 WT ($n$ = 8), L438K ($n$ = 8), and A439K ($n$ = 6) (**Figure 4—source data 1**). Currents were elicited by the voltage protocol shown next to the listed pressures. Dotted lines denote zero current. (**f**) Quantification of half-maximal voltage at −50 mmHg for WT (109 mV), L438K (67 mV), and A439K (63 mV) (**Figure 4—source data 1**). Statistical comparison was conducted with unpaired $t$-tests with Welch's correction (***p < 0.001, ****p < 0.0001). (**g**) Quantification of activation $\tau_{on}$ at −50 mmHg and 160 mV for WT (41 ms), L438K (13 ms), and A439K (16 ms) (**Figure 4—source data 1**). Statistical comparison was conducted with unpaired $t$-tests with Welch's correction (***p < 0.001, ****p < 0.0001). (**h**) A lysine mutation along TM 4 converts the OSCA1.2 channel into a phospholipid scramblase with spontaneous phospholipid permeability. All error bars represent standard error of the mean (SEM) calculated from replicate images or independent patches.

The online version of this article includes the following source data and figure supplement(s) for figure 4:

**Source data 1.** Results from the lysine mutations on TM 4 of OSCA1.2.

*Figure 4 continued on next page*

Figure 4 continued

**Figure supplement 1.** Atomistic molecular dynamics (MD) simulations of OSCA1.2 wildtype (WT) and L438K exhibit differential hydration of the pore region.

**Figure supplement 1—source data 1.** Results from molecular dynamics (MD) simulation of OSCA1.2.

**Figure supplement 2.** Role of K435 in OSCA1.2 mutant phospholipid permeability.

**Figure supplement 2—source data 1.** Results from OSCA1.2 K435 and L438 mutations.

**Figure supplement 3.** Hypotonic stimulation activates the OSCA1.2 ion channel.

**Figure supplement 3—source data 1.** Results from hypotonic stimulation of OSCA1.2.

hydrophilic side chains within the hydrophobic core of the membrane promote trans-bilayer phospholipid transport and this effect is further enhanced when these hydrophilic residues are stacked within the helix (*Nakao and Nakano, 2022*). To explore the role of the endogenous K435 in facilitating L438K-mediated spontaneous phospholipid permeability, we assessed PS exposure in the K435L and K435L/L438K mutants. Remarkably, the double mutant K435L/L438K, which shifts the lysine to position 438, resulted in a modest level of spontaneous PS exposure (31%, *Figure 4—figure supplement 2b, c*). This level is intermediate between the WT (K435) with no PS exposure and the L438K mutant (K435/K438) with 63% spontaneous PS exposure (*Figure 4c*; *Figure 4—figure supplement 2b, c*). In contrast, K435L (L435/L438) did not induce significant spontaneous PS exposure (3%, *Figure 4—figure supplement 2b, c*). These findings, consistent with previous studies on model transmembrane peptides (*Nakao and Nakano, 2022*), highlight the role of multiple charged resides in TM 4 in promoting phospholipid scrambling. Moreover, our results further support the hypothesis that the instability of the TM 4/6 interface in OSCA1.2 is crucial for controlling phospholipid and ion permeation.

## A439K on TM4 converts OSCA1.2 into an osmolarity-sensing scramblase

Interestingly, mutating the neighboring amino acid (*Figure 4a*), A439, to lysine resulted in minimal spontaneous PS exposure (10%, *Figure 4b, c*). We thus reasoned that A439K scramblase activity may require additional stimulation to trigger scramblase activity. Given that WT OSCA1.2 ion channel activity can be induced by hypotonic treatment (*Figure 4—figure supplement 3*), we acutely treated WT- and A439K-expressing cells with a hypotonic solution (120 mOsm/kg) and assessed scramblase activity using time-lapse imaging. Indeed, PS exposure was robustly induced for the A439K mutant (*Figure 5b, c*) but not WT (*Figure 5a, c*) in response to hypotonic stimulation. Inside-out patch clamp further demonstrated that A439K enhances OSCA1.2 ion channel activity as evidenced by the accelerated activation kinetics (*Figure 4g*) and left-shifted *G–V* relationship (*Figure 4f*). Our experiments thus indicate that A439K modestly disrupts OSCA1.2 gating and converts the osmolarity-activated ion channel (*Figure 4—figure supplement 3*) into an osmolarity-sensing phospholipid scramblase (*Figure 5d*).

## TM 4 lysine mutations convert TMEM63A into a constitutively active scramblase

Finally, we turned our attention to TMEM63A to further investigate the evolutionary conservation of our observation that TM 4 lysine mutations convert TMEM16 and OSCA members into scramblases. TMEM63s represent the mammalian members of the OSCA/TMEM63 family with three members present in humans (TMEM63A–C). Recent structural and functional characterizations indicate that TMEM63s function as mechanosensitive ion channels gated by high-threshold membrane tension and, in notable contrast to all other structurally resolved TCS superfamily members, they likely function as monomers (*Zheng et al., 2023*; *Qin et al., 2023*; *Wu et al., 2024*). Given their structural homology to TMEM16s and OSCA1.2 (*Zheng et al., 2023*), we hypothesized that TM 4 mutations in TMEM63A would also result in GOF activity. We again identified residues near the putative gate by structural (*Figure 6a*) and sequence alignment (*Figure 1—figure supplement 1f*), selecting W472 (equivalent to F518 in TMEM16F, L543 in TMEM16A, and K435 in OSCA1.2), S475 (equivalent to I521 in TMEM16F, I546 in TMEM16A, and L438 in OSCA1.2), and A476 (equivalent to M522 in TMEM16F, I547 in TMEM16A, and A439 in OSCA1.2). Indeed, overexpressing eGFP-tagged mouse TMEM63A

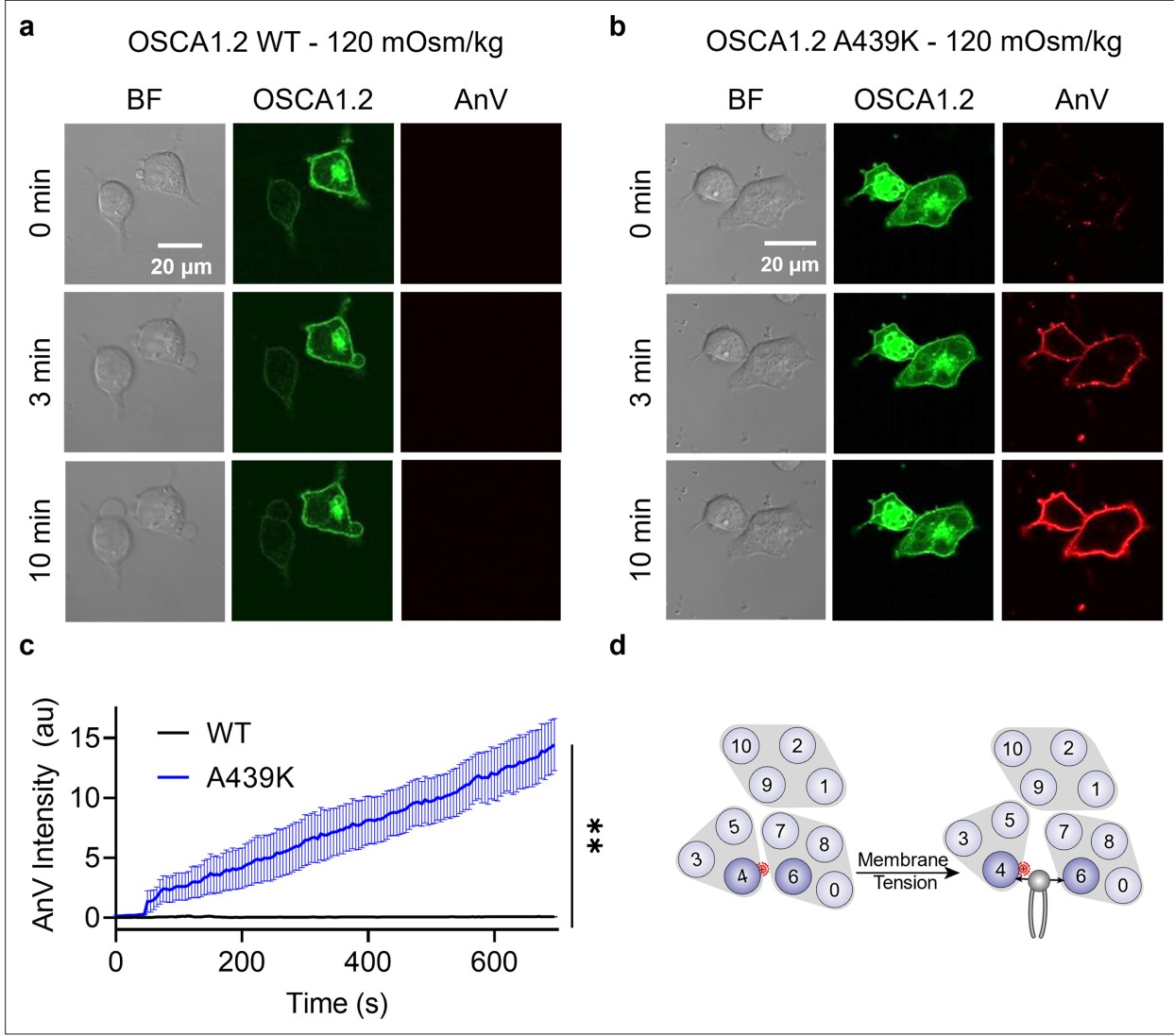

**Figure 5.** OSCA1.2 A439K is an osmolarity-activated scramblase. Representative images of hypotonic osmolarity stimulation of TMEM16F knockout (KO) HEK293T cells expressing eGFP-tagged OSCA1.2 (**a**) wildtype (WT) or (**b**) the A439K mutant (center columns). CF 594-conjugated Annexin V (AnV) (right columns) labeled phosphatidylserine (PS) exposing cells. BF denotes bright field images (left columns). Each row of representative images corresponds to the indicated time after hypo-osmotic stimulation. (**c**) Quantification of AnV intensity for OSCA1.2 WT (*n* = 5) and A439K (*n* = 5) after hypo-osmotic stimulation (*Figure 5—source data 1*). Statistical comparison was conducted with an unpaired *t*-test with Welch's correction (**p < 0.01). (**d**) The A439K mutation converts OSCA1.2 to an osmolarity-activated phospholipid scramblase. Error bars represent standard error of the mean (SEM).

The online version of this article includes the following source data for figure 5:

**Source data 1.** Results from osmolarity activation of OSCA1.2 A439K scramblase.

with single lysine mutations at either W472 or S475 led to spontaneous PS exposure (*Figure 6b, c*), though the AnV staining revealed punctate rather than global patterns of PS exposure induced by their OSCA1.2 counterparts. Notably, A476K failed to show obvious PS exposure (*Figure 6b, c*), even after 120 mOsm/kg hypotonic treatment (*Figure 6—figure supplement 1b, c*), perhaps reflecting differences in mechanical pressure threshold between OSCA1.2 and TMEM63A (*Zheng et al., 2023*). To confirm membrane localization and further probe mutant effects, we exploited TMEM63A ion channel function using the cell-attached patch clamp configuration. As TMEM63A exhibits voltage-dependent activity under high pressure, we compared mutant and WT *I–V* relationships at −80 mmHg. W472K, S475K, and A476K caused marked reductions in $V_{0.5}$ from 122.3 ± 3.5 mV for WT to 95.9 ± 6.3, 92.1 ± 8.6, and 96.8 ± 4.2 mV, respectively (*Figure 6d–f*). By comparison, mock-transfected controls failed to elicit current (*Figure 6d, e*). Together, these results are consistent with our observations in TMEM16F,

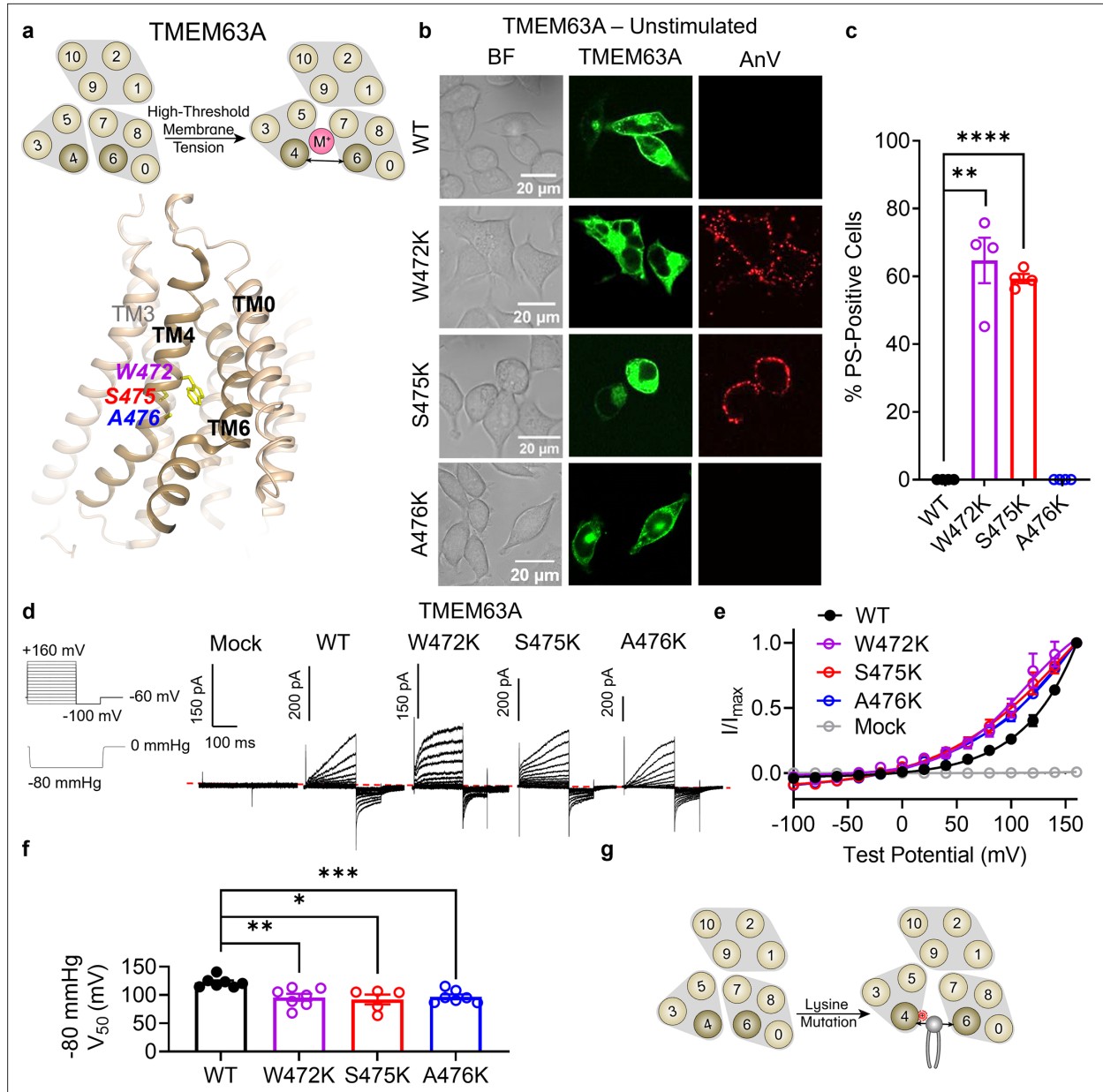

**Figure 6.** Lysine mutations along TM 4 enable TMEM63A channel and scramblase activities. (**a**) Top: TMEM63A is an ion channel gated by high-threshold membrane tension. Bottom: the TM 4/6 interface of *Hs*TMEM63A (PDB 8GRS) with key residues shown as yellow sticks using amino acid numbering corresponding to the mouse ortholog. (**b**) Representative images of TMEM16F knockout (KO) HEK293T cells expressing eGFP-tagged TMEM63A wildtype (WT), W472K, S475K, and A476K (center column). CF 594-conjugated Annexin V (AnV) (right column) labeled phosphatidylserine (PS) exposing cells. BF denotes bright field images (left column). (**c**) Quantification of the percentage of cells with AnV labeling for TMEM63A WT ($n = 4$), W472K ($n = 4$), S475K ($n = 4$), and A476K-transfected cells ($n = 4$) (*Figure 6—source data 1*). Statistical comparisons were conducted with unpaired $t$-tests with Welch's correction (**$p < 0.01$, ****$p < 0.0001$). (**d**) Representative current recordings and (**e**) normalized conductance–voltage (*I–V*) relationships of cell-attached patches from TMEM16F KO HEK293T cells expressing either eGFP mock-transfected ($n = 7$) or eGFP-tagged TMEM63A WT ($n = 7$), W472K ($n = 7$), S475K ($n = 5$), or A476K ($n = 7$) (*Figure 6—source data 1*). Currents represent the subtraction of voltage alone from currents elicited by the voltage and pressure protocols shown. Dotted line denotes zero current. Note that the mock control was normalized to the mean maximal current elicited from WT-transfected cells. (**f**) Quantification of half-maximal voltage at −80 mmHg for WT (122 mV), W472K (96 mV), S475K (92 mV), and A476K (97 mV) (*Figure 6—source data 1*). Statistical comparison was conducted with unpaired $t$-tests with Welch's correction (*$p < 0.05$, **$p < 0.01$, ***$p < 0.001$). (**g**) Lysine mutations along TM 4 in TMEM63A enable spontaneous phospholipid permeability. All error bars represent standard error of the mean (SEM) calculated from replicate images or independent patches.

The online version of this article includes the following source data and figure supplement(s) for figure 6:

**Source data 1.** Results from the lysine mutations on TM 4 of TMEM63A.

*Figure 6 continued on next page*

*Figure 6 continued*

**Figure supplement 1.** TMEM63A hypotonic stimulation.

**Figure supplement 1—source data 1.** Results from TMEM63A hypotonic stimulation.

TMEM16A, and OSCA1.2, indicating that single lysine mutations along TM 4 of TMEM63A can facilitate both ion and phospholipid permeation (*Figure 6g*).

## Discussion

Mechanistically, our study improves models of TMEM16 substrate permeation and gating. We identified multiple mutations in TMEM16F and TMEM16A that promote phospholipid permeation and cause commensurate changes in ion channel activities. Our findings complement functional characterizations of TMEM16F gating mutants where gate destabilization is inversely correlated with side chain hydropathy (*Le et al., 2019a*; *Arndt et al., 2022*). Lysine is above only arginine on the hydropathy index and thus likely explains why it readily destabilizes the gate. For instance, F518K exhibits spontaneous PS exposure, even when the calcium-binding site is destroyed (*Le et al., 2019a*), whereas F518H does not exhibit spontaneous PS exposure (*Arndt et al., 2022*). Interestingly, the recent TMEM16F F518H structure (*Figure 1—figure supplement 1e*) shows local membrane thinning due in part to unexpected conformational changes in TM 3 (*Arndt et al., 2022*). Future structural studies are needed to assess whether TMEM16A or OSCA/TMEM63 mutant scramblases also promote membrane thinning and/or conformational rearrangements in TM 3. More broadly, our results highlight an increasingly appreciated design principle of scramblases where polar and charged residues often line a membrane-spanning groove. This observation has been noted for TMEM16 (*Bushell et al., 2019*; *Brunner et al., 2014*; *Alvadia et al., 2019*), Xkr (*Straub et al., 2021*; *Sakuragi et al., 2021*), and opsin (*Morra et al., 2018*) scramblases and should be a key criterion for identifying and characterizing new scramblases.

Our findings also advance our understanding of evolutionary relatives of the TCS superfamily (*Le et al., 2021*) and help uncover their roles in human diseases. Although OSCA/TMEM63 proteins are not known to scramble phospholipids and we did not detect obvious hypotonicity-induced PS exposure in cells overexpressing WT OSCA1.2 (*Figure 5a*) or TMEM63A (*Figure 1—figure supplement 1a*), we show that single mutations in TM 4 of OSCA1.2 and TMEM63A convert these osmolarity-activated and/or mechanosensitive ion channels into phospholipid scramblases, similar to our findings with TMEM16A mutants. We thus speculate that the conserved structural architecture in the transmembrane region endows TCS proteins with a potential to scramble phospholipids, though this capability may have been lost by some members during evolution. This hypothesis is especially intriguing given the recent OSCA open state structures, which detail dramatic conformational rearrangements of TMs 3–6 leading to a phospholipid-lined (or proteolipidic) pore near TMs 4 and 6 (*Han et al., 2024*). It will be interesting to test whether equivalent mutations can convert transmembrane channel-like (TMC) proteins—the third TCS relative of TMEM16 and OSCA/TMEM63 (*Le et al., 2021*)—into phospholipid scramblases. TMC1 is best known for its role in auditory sensation, and thus far has mostly been characterized in vivo due to expression difficulties in heterologous systems (*Pan et al., 2018*). However, recent in vivo characterization of mouse TMC1 M412K, known as the *Beethoven* mutation, provided an important insight (*Ballesteros and Swartz, 2022*). The deafness-associated mutation is located in TM 4 (*Figure 1—figure supplement 1f*; equivalent of M522 in TMEM16F, I547 in TMEM16A, A439 in OSCA1.2, and A476 in TMEM63A) and results in constitutive PS exposure when expressed in the hair cell membranes of both heterozygous and homozygous mice (*Ballesteros and Swartz, 2022*). This raises the intriguing possibility that the *Beethoven* mutation may enable TMC1 to spontaneously permeate phospholipids, leading to a loss of membrane homeostasis and ultimately, auditory sensation (*Sakuragi and Nagata, 2023*). The possibility of converting TMC proteins into phospholipid scramblases should be thoroughly investigated. Additionally, disease-associated mutations in TMEM63 proteins are present along the TM 4/6 interface, such as TMEM63B T481N (*Zheng et al., 2023*). We speculate that introducing more hydropathic side chains along this interface may lead to spontaneous ion and or phospholipid permeability, perhaps contributing to underlying pathophysiology.

# Materials and methods

## Key resources table

| Reagent type (species) or resource | Designation | Source or reference | Identifiers | Additional information |
|---|---|---|---|---|
| Gene (*Mus musculus*) | MmTMEM16F | NCBI | NP_780553.2 | Contains a three amino acid N-terminal truncation (MQM) |
| Gene (*M. musculus*) | MmTMEM16A | NCBI | NP_001229278 | |
| Gene (*Arabidopsis thaliana*) | AtOSCA1.2 | GeneBank | AIU34614.1 | |
| Gene (*M. musculus*) | MmTMEM63A | NCBI | NP_001404481.1 | |
| Commercial assay or kit | In-Fusion Snap Assembly | Takara | 638947 | |
| Recombinant DNA reagent | peGFP-N1 | Addgene: Vector Database—pEGFP-N1 | | |
| Recombinant DNA reagent | peGFP-N1_MmTMEM16F | *Le et al., 2019a* | | |
| Recombinant DNA reagent | peGFP-N1_MmTMEM16A | *Le et al., 2019c* | | |
| Recombinant DNA reagent | peGFP-N1_AtOSCA1_2 | This paper | | Subcloned with In-fusion (IF) primers |
| Recombinant DNA reagent | peGFP-N1_MmTMEM63A | This paper | | Subcloned with In-fusion (IF) primers |
| Sequence-based reagent | eGFP-N1_678–679_F | This paper | PCR primers | ATGGTGAGCAAGGGCGAGG |
| Sequence-based reagent | eGFP-N1_678–679_R | This paper | PCR primers | GGTGGCGACCGGTGGATCC |
| Sequence-based reagent | IFpeGFP-N1_OSCA1_2-eGFP_F | This paper | PCR primers | CCACCGGTCGCCACCATGGCGACACTTCAGGAT |
| Sequence-based reagent | IFpeGFP-N1_OSCA1_2-eGFP_R | This paper | PCR primers | GCCCTTGCTCACCATGACTAGTTTACCACTAAAGGGC |
| Sequence-based reagent | IFpeGFP-N1_MmTMEM63A-eGFP_F | This paper | PCR primers | CCACCGGTCGCCACCATGACCAGCTCCCCGTTCC |
| Sequence-based reagent | IFpeGFP-N1_MmTMEM63A-eGFP_R | This paper | PCR primers | GCCCTTGCTCACCATGGATTCCTGGTAGGCATAAGC |
| Sequence-based reagent | MmTMEM16F_I521K_F | This paper | PCR primers | TTCATCATCAAGATGATCCTCAACACG |
| Sequence-based reagent | MmTMEM16F_I521K_R | This paper | PCR primers | GAGGATCATCTTGATGATGAAGCTGATG |
| Sequence-based reagent | MmTMEM16F_I521A_F | This paper | PCR primers | CTTCATCATCGCCATGATCCTCAACAC |
| Sequence-based reagent | MmTMEM16F_I521A_R | This paper | PCR primers | GAGGATCATGGCGATGATGAAGCTG |
| Sequence-based reagent | MmTMEM16F_I521E_F | This paper | PCR primers | TCATCATCGAGATGATCCTCAACACG |
| Sequence-based reagent | MmTMEM16F_I521E_R | This paper | PCR primers | TGAGGATCATCTCGATGATGAAGCTG |
| Sequence-based reagent | MmTMEM16F_M522K_F | This paper | PCR primers | ATCATCATCAAGATCCTCAACACG |
| Sequence-based reagent | MmTMEM16F_M522K_R | This paper | PCR primers | TGAGGATCTTGATGATGATGAAGC |
| Sequence-based reagent | MmTMEM16F_T526K_F | This paper | PCR primers | TCCTCAACAAGATCTACGAGAAGGTGG |
| Sequence-based reagent | MmTMEM16F_T526K_R | This paper | PCR primers | CTCGTAGATCTTGTTGAGGATCATGATG |
| Sequence-based reagent | MmTMEM16F_I611K_F | This paper | PCR primers | GCTGACGAAGATCATGGGGGGA |

*Continued on next page*

*Continued*

| Reagent type (species) or resource | Designation | Source or reference | Identifiers | Additional information |
|---|---|---|---|---|
| Sequence-based reagent | MmTMEM16F_I611K_R | This paper | PCR primers | CCCCCATGATCTTCGTCAGCTGTG |
| Sequence-based reagent | MmTMEM16A_I546K_F | This paper | PCR primers | CGTGGTCAAGATTCTGCTGGATGAAG |
| Sequence-based reagent | MmTMEM16A_I546K_R | This paper | PCR primers | CCAGCAGAATCTTGACCACGAGGT |
| Sequence-based reagent | MmTMEM16A_I547K_F | This paper | PCR primers | TGGTCATCAAGCTGCTGGATGAAG |
| Sequence-based reagent | MmTMEM16A_I547K_R | This paper | PCR primers | TCCAGCAGCTTGATGACCACGAG |
| Sequence-based reagent | MmTMEM16A_E551K_F | This paper | PCR primers | TCTGCTGGATAAGGTTTACGGCTGC |
| Sequence-based reagent | MmTMEM16A_E551K_R | This paper | PCR primers | GCCGTAAACCTTATCCAGCAGAATGATG |
| Sequence-based reagent | AtOSCA1_2_K435L_F | This paper | PCR primers | TTGCACTGCTGCTTTTCCTCGC |
| Sequence-based reagent | AtOSCA1_2_K435L_R | This paper | PCR primers | GAGGAAAAGCAGCAGTGCAATACCC |
| Sequence-based reagent | AtOSCA1_2_K435L_L438K_F | This paper | PCR primers | ATTGCACTGCTGCTTTTCAAGGCC |
| Sequence-based reagent | AtOSCA1_2_K435L_L438K_R | This paper | PCR primers | CTTGAAAAGCAGCAGTGCAATACCCG |
| Sequence-based reagent | AtOSCA1_2_L438K_F | This paper | PCR primers | AAGCTTTTCAAGGCC TTTCTGCCATC |
| Sequence-based reagent | AtOSCA1_2_L438K_R | This paper | PCR primers | CAGAAAGGCCTTGA AAAGCTTCAGTGC |
| Sequence-based reagent | AtOSCA1_2_A439K_F | This paper | PCR primers | GCTTTTCCTCAAGT TTCTGCCATCC |
| Sequence-based reagent | AtOSCA1_2_A439K_R | This paper | PCR primers | GGCAGAAACTTGAGGAAAAGCTTCAG |
| Sequence-based reagent | MmTMEM63A_W472K_F | This paper | PCR primers | CTCCTGCTGAAGTCCTTCTCTGCG |
| Sequence-based reagent | MmTMEM63A_W472K_R | This paper | PCR primers | AGAGAAGGACTTCAGCAGGAGTGTGG |
| Sequence-based reagent | MmTMEM63A_S475K_F | This paper | PCR primers | GGTCCTTCAAGGCGCTGCTTCCG |
| Sequence-based reagent | MmTMEM63A_S475K_R | This paper | PCR primers | AAGCAGCGCCTTGAAGGACCACAGC |
| Sequence-based reagent | MmTMEM63A_A476K_F | This paper | PCR primers | TCCTTCTCTAAGCT GCTTCCGTCCA |
| Sequence-based reagent | MmTMEM63A_A476K_R | This paper | PCR primers | CGGAAGCAGCTT AGAGAAGGACCAC |
| Cell line (*Homo sapiens*) | TMEM16F KO HEK293T | **Le et al., 2019b** | | |
| Peptide, recombinant protein | AnnexinV-594 | Biotium | #29011 | |
| Chemical compound, drug | Ani9 | Sigma-Aldrich | SML1813 | |
| Software, algorithm | Scrambling assay | https://github.com/yanghuanghe/scrambling_activity, copy archived at **Huanghe, 2020** | | |

## Cloning and mutagenesis

All constructs used a peGFP-N1 vector backbone. GFP mock controls used the empty peGFP-N1 vector. WT sequence and mutation numbers correspond to NCBI: NP_780553.2 (*Mus musculus* TMEM16F) with a three amino acid (MQM) N-terminal truncation, NCBI: NP_001229278 (*M. musculus*

TMEM16A), GenBank: AIU34614.1 (*A. thaliana* OSCA1.2), and NCBI: NP_001404481.1 (*M. musculus* TMEM63A). *At*OSCA1.2 and *Mm*TMEM63A cDNAs were subcloned using In-Fusion Snap Assembly (Takara Bio, Kusatsu, Japan, Cat. #638947). Point mutants were generated by PCR site-directed mutagenesis with primers from IDT DNA Technologies (Coralville, IA). Sequences were confirmed by Sanger sequencing (Azenta, Burlington, MA).

## Bioinformatics

The following sequences were obtained from UniProt and aligned using Clustal Omega: Q8BHY3-2 (*Mm*TMEM16A), Q9NQ90 (*Hs*TMEM16B), Q9BYT9 (*Hs*TMEM16C), Q32M45 (*Hs*TMEM16D), Q75V66 (*Hs*TMEM16E), Q6P9J9 (*Mm*TMEM16F), Q6IWH7 (*Hs*TMEM16G), Q9HCE9 (*Hs*TMEM16H), A1A5B4 (*Hs*TMEM16J), Q9NW15 (*Hs*TMEM16K), C7Z7K1 (*Nh*TMEM16), Q4WA18 (*Af*TMEM16), Q9XEA1 (*At*OSCA1.1), Q5XEZ5 (*At*OSC1.2), B5TYT3 (*At*OSCA1.3), A0A097NUQ0 (*At*OSCA1.4), A0A097NUS0 (*At*OSCA1.5), A0A097NUP1 (*At*OSCA1.6), A0A097NUP8 (*At*OSCA1.7), A0A097NUQ2 (*At*OSCA1.8), A0A097NUQ5 (*At*OSCA2.1), A0A097NUS5 (*At*OSCA2.2), A0A097NUP6 (*At*OSCA2.3), A0A097NUQ3 (*At*OSCA2.4), A0A097NUQ7 (*At*OSCA2.5), Q9C8G5 (*At*OSCA3.1), A0A097NUT0 (*At*OSCA4.1), Q91YT8 (*Mm*TMEM63A), Q5T3F8 (*Hs*TMEM63B), and Q9P1W3 (*Hs*TMEM63C), Q8R4P5 (*Mm*TMC1), Q8R4P4 (*Mm*TMC2), Q7TQ69 (*Mm*TMC3), Q7TQ65 (*Mm*TMC4), Q32NZ6 (*Mm*TMC5), Q7TN60 (*Mm*TMC6), Q8C428 (*Mm*TMC7), and Q7TN58 (*Mm*TMC8). A subset of the alignment was selected for *Figure 1—figure supplement 1*. Structural models were obtained from the PDB, aligned, and visualized using Pymol (Schrödinger, New York, NY).

## Cell culture

The HEK293T cell line was authenticated and tested negative of mycoplasma contamination by the Duke Life Science Facility. The TMEM16F KO HEK293T cell line was generated by the Duke Functional Genomics Core and characterized in previous studies (*Le et al., 2019b*; *Le et al., 2019a*). All cells were cultured with DMEM (Dulbecco's Modified Eagle Medium) (Gibco/Thermo Fisher Scientific, Waltham, MA, Cat. #11995-065) supplemented with 10% fetal bovine serum (Sigma-Aldrich, St. Louis, MO, Cat. # F2442) and 1% penicillin/streptomycin (Gibco/Thermo Fisher Scientific, Waltham, MA, Cat. #15-140-122) at 37°C in 5% $CO_2$-95% air. Cells tested negative for mycoplasma contamination (Applied Biological Materials, Vancouver, Canada, Cat. #G239).

## Transfection

Plasmids were transiently transfected into TMEM16F KO HEK 293T cells by using X-tremeGENE9 (MilliporeSigma, Burlington, MA, Cat. #XTG9-RO), X-tremeGENE360 (MilliporeSigma, Burlington, MA, Cat. #XTG360-RO), or Lipofectamine 2000 (Thermo Fisher Scientific, Waltham, MA, Cat. # 11668027). Media was replaced with calcium-free DMEM (Gibco/Thermo Fisher Scientific, Waltham, MA, Cat. #21068-028) 3–4 hr after transfection. The transfected cells were imaged or patched 18–24 or 18–48 hr after transfection, respectively.

## Fluorescence imaging of scramblase-mediated PS exposure

A Zeiss 780 inverted confocal microscope was used to monitor scramblase activity in live cells using the methods described in previous publications (*Zhang et al., 2020*; *Le et al., 2019b*; *Le et al., 2019a*). 18–24 hr after transfection, the cells were incubated in AnV buffer (1:175 dilution of the fluorescently tagged AnV (Biotium, Fremont, CA, Cat. #29011) in Hank's balanced salt solution) immediately before and throughout the duration of the imaging experiment without a formal incubation period. For *Figure 3—figure supplement 1*, 10 μM Ani9 (Sigma-Aldrich, St. Louis, MO, Cat. #SML1813) was added into the AnV solution before application. Spontaneous PS-positive cells were readily labeled by fluorescently tagged AnV. Results were quantified as a percentage of PS-positive cells among all GFP-positive cells. For osmolarity activation, 2 mM $CaCl_2$ in $ddH_2O$ was added to the AnV buffer at a 2:1 ratio. The final osmolarity was ~120 mOsm/kg as measured by a micro-osmometer (Advanced Instrument, Norwood, MA). Cells overexpressing WT or mutant were treated with low osmolarity AnV buffer, and the scramblase activity was measured by recording fluorescent AnV surface accumulation at 5- (*Figure 5*) or 7- (*Figure 6—figure supplement 1*) s intervals. A custom MATLAB code was used to quantify AnV signal and is available at GitHub (https://github.com/yanghuanghe/scrambling_activity, copy archived at *Huanghe, 2020*; *Le et al., 2019a*).

## Electrophysiology

All electrophysiology recordings were conducted using an Axopatch 200B amplifier with the signal digitally sampled at 10 kHz using an Axon Digidata 1550A (Molecular Devices, Inc, San Jose, CA). All electrophysiology recordings were carried out at room temperature 18–48 hr after transfection. Glass pipettes were pulled from borosilicate capillaries (Sutter Instruments, Novato, CA) and fire-polished using a microforge (Narishige, Amityville, NY). Pipettes had resistances of 2–4 MΩ in the physiological bath solution.

### Inside-out patch clamp recordings

The pipette solution (external) contained 140 mM NaCl, 10 mM HEPES (Hydroxyethylpiperazine Ethane Sulfonic Acid), 2 mM MgCl$_2$, adjusted to pH 7.3 (NaOH), and the bath solution (internal) contained 140 mM NaCl, 10 mM HEPES, 5 mM EGTA, adjusted to pH 7.3 (NaOH). OSCA1.2 WT and mutations were held at constant pressure administrated using a syringe calibrated with a manometer, similar to a previous study (*Tsuchiya et al., 2018*). In our experience, the constant pressure and variable voltage protocol achieved more consistent measurements. Patches were held at a membrane potential of −60 mV and at the indicated pressure, then stimulated using the indicated voltage protocol, taking advantage of the larger OSCA1.2 currents elicited by depolarizing potentials.

### Cell-attached patch clamp recordings

For TMEM63 recordings, cell-attach mode was used instead of inside-out to avoid breaking the giga-ohm seal under higher applied pressure and voltage. The bath solution contained (in mM): 140 KCl, 10 HEPES, 2 MgCl$_2$, 10 glucose, pH 7.3 adjusted with KOH. The pipette solution contained (in mM): 130 NaCl, 5 KCl, 10 HEPES, 10 TEA-Cl (Tetraethylammonium chloride), 1 CaCl$_2$, 1 MgCl$_2$, pH 7.3 (with NaOH). The mechano-activated current was evoked with a 200-ms pressure pulse at −80 mmHg using a high-speed pressure clamp system (ALA Scientific Instruments, Farmingdale, NY, Cat. #HSPC-1). The membrane potential inside the patch was held at −60 mV. The voltage pulse alone was run first followed by voltage pulse with pressure. The mechanosensitive current was obtained by subtracting the voltage pulse from the voltage pulse with pressure.

### Whole-cell TMEM16 patch clamp recordings

The pipette solution (internal) contained 140 mM CsCl, 10 mM HEPES, 5 mM EGTA, adjusted to pH 7.3 (CsOH), and the bath solution (external) contained 140 mM NaCl, 10 mM HEPES, 5 mM EGTA, adjusted to pH 7.3 (NaOH). Patches were held at a membrane potential of −60 mV, then stimulated using the indicated voltage protocol. For *Figure 3—figure supplement 1*, 10 µM Ani9 (Sigma-Aldrich, St. Louis, MO, Cat. #SML1813) was added to bath solution and applied using a perfusion manifold (ALA Scientific Instruments, Cat. #ALA-VM8).

### Whole-cell hypotonic patch clamp recordings

The pipette solution (internal) contained 140 mM Na gluconate, 10 mM HEPES, 1 mM MgCl$_2$, and 0.2 mM EGTA, adjusted to pH 7.3 (NaOH). Cells seeded on a small section of cover glass were placed in the bath solution (external) containing 140 mM Na gluconate, 10 mM HEPES, 1 mM MgCl$_2$, and 0.2 mM EGTA adjusted to pH 7.3 (NaOH). Patches were held at a membrane potential of −60 mV, then stimulated using the indicated voltage protocol using a 2-s sweep protocol. After five sweeps, ddH$_2$O was slowly and gently added to the bath at a ratio of 2:1 using a hypodermic needle to induce hypotonic cell swelling. The sweep protocol continued for a minimum of 4 min after hypotonic stimulation.

## Data analysis for electrophysiology

All data analysis was performed using Clampfit (Molecular Devices, Inc, San Jose, CA), Excel (Microsoft, Redmond, WA), MATLAB (MathWorks, Portola Valley, CA), and Prism software (GraphPad, Boston, MA). Individual *G–V* curves were fitted with a Boltzmann function,

$$G = \frac{G_{max}}{1 + e^{\frac{-ZF\left(V - V_{0.5}\right)}{RT}}} \tag{1}$$

where $G_{max}$ denotes the fitted value for maximal conductance, $V_{0.5}$ denotes the voltage of half-maximal activation of conductance, $Z$ denotes the net charge moved across the membrane during the transition from the closed to the open state, and $F$ denotes the Faraday constant.

## Model building and MD simulations

The initial model of OSCA1.2 for atomistic MD simulations and homology modeling was constructed based on PDB: 6MGV. Missing segments were modeled using the Swiss-PDB server (https://swiss-model.expasy.org/). The OSCA1.2 L438K mutation structure was modeled using 6MGV as the template, and the RMSD between the mutation and the original structure of 6MGV was less than 0.5 Å. To mimic the state of transmembrane proteins in the plasma membrane, the model was inserted into a lipid membrane consisting of POPC (Palmitoyl oleoyl phosphatidyl choline) (exoplasmic leaflet) and 1:1 POPS/POPE (Palmitoyl oleoyl phosphatidylserine/palmitoyl oleoyl phosphatidylethanolamine) (cytoplasmic leaflet) using the CHARMM-GUI web server (*Jo et al., 2008*). To neutralize charge levels in the simulation system, 150 mM KCl solution was added. The final simulation boxes contained ~600 lipid molecules, with ~290,000 atoms. During the simulation, the CHARMM36m all-atom force field was used. The temperature was maintained at 310 K to maintain the fluidity of lipids. During the atomistic simulation, long-range electrostatic interactions were described by the Particle Mesh Ewald algorithm with a cutoff of 12 Å. Van der Waals interactions were cut off at 12 Å and the MD time step was set at 2 fs. All atomistic MD simulations were conducted using GROMACS-2022.3 software package and enhanced sampling MD simulations were conducted using GROMACS-2022.3 with the COLVARS module. Before the production process, the system performed 5000 steps of energy minimization. Subsequently, all systems were equilibrated using the default equilibration parameters provided by CHARMM-GUI. After equilibration, the velocities of all atoms were randomly generated, and the product simulation system consisted of a 2-stage MD simulation. In step 1, the system used a conventional MD simulation run for 200 ns. In step 2, the system used metadynamics simulations run for ~700 ns. During enhanced sampling MD simulations, the sampling interval of the collective variable was adjusted to adapt the lipid movement trajectory. VMD and PYMOL were used to analyze simulation trajectories and water occupancy.

## Materials availability

All plasmids generated in this study can be requested from the corresponding authors.

## Acknowledgements

This work was supported by the National Institutes of Health (DP2GM126898, R21GM146152, R35GM153196, and Duke Science & Technology SPARK Award to HY) and the National Science Foundation Graduate Research Fellowship Program (DGE 2139754 to AJL). Any opinions, findings, and conclusions or recommendations expressed in this material are those of the author(s) and do not necessarily reflect the views of the National Science Foundation.

## Additional information

### Funding

| Funder | Grant reference number | Author |
| --- | --- | --- |
| National Institute of General Medical Sciences | R21GM146152 | Huanghe Yang |
| National Institute of General Medical Sciences | DP2GM126898 | Huanghe Yang |
| National Institute of General Medical Sciences | R35GM153196 | Huanghe Yang |
| Duke University | Science & Technology SPARK Award | Huanghe Yang |

| Funder | Grant reference number | Author |
| --- | --- | --- |
| National Science Foundation | DGE 2139754 | Augustus J Lowry |

The funders had no role in study design, data collection, and interpretation, or the decision to submit the work for publication.

## Author contributions

Augustus J Lowry, Data curation, Formal analysis, Validation, Investigation, Visualization, Methodology, Writing – original draft, Writing – review and editing; Pengfei Liang, Data curation, Formal analysis, Validation, Investigation, Visualization, Methodology; Mo Song, Data curation, Software, Formal analysis, Visualization, Methodology; Yuichun Wan, Investigation; Zhen-Ming Pei, Resources; Huanghe Yang, Conceptualization, Resources, Data curation, Supervision, Funding acquisition, Visualization, Methodology, Writing – original draft, Project administration, Writing – review and editing; Yang Zhang, Conceptualization, Data curation, Software, Formal analysis, Investigation, Methodology, Writing – review and editing

## Author ORCIDs

Augustus J Lowry ⓘ https://orcid.org/0000-0002-3524-3008
Huanghe Yang ⓘ https://orcid.org/0000-0001-9521-9328
Yang Zhang ⓘ https://orcid.org/0000-0003-3625-9965

Reviewer #1 (Public review): https://doi.org/10.7554/eLife.96957.3.sa1
Reviewer #2 (Public review): https://doi.org/10.7554/eLife.96957.3.sa2
Reviewer #3 (Public review): https://doi.org/10.7554/eLife.96957.3.sa3
Author response https://doi.org/10.7554/eLife.96957.3.sa4

# Additional files

## Supplementary files

• MDAR checklist

## Data availability

A custom MATLAB code to quantify AnV signal is available at GitHub, copy archived at *Huanghe, 2020*. All other data are included in the manuscript and/or Supplementary files. Source data files have been provided for Figures 1–6.

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
