## [Editor Report · eLife Assessment]

This **important** study advances our understanding of the mechanisms controlling lipid flux and ion permeation in the TMEM16 and OSCA/TMEM63 family channels. The study provides **compelling** new evidence indicating that side chains along the TM4/6 interface play a key role in gating lipid and ion fluxes in these channels. The authors suggest that the transmembrane channel/scramblase family proteins may have originally functioned as scramblases but lost this capacity over evolution.

---

## [Referee Report · Reviewer #1 (Public review)]

Summary:

TMEM16, OSCA/TMEM63, and TMC belong to a large superfamily of ion channels where TMEM16 members are calcium activated lipid scramblases and chloride channels, whereas OSCA/TMEM63 and TMCs are mechanically activated ion channels. In the TMEM16 family, TMEM16F is a well characterized calcium activated lipid scramblase that play an important role in processes like blood coagulation, cell death signaling, and phagocytosis. In a previous study the group has demonstrated that lysine mutation in TM4 of TMEM16A can enable the calcium activated chloride channel to permeate phospholipids too. Based on this they hypothesize that the energy barrier for lipid scramblase in these ion channels is low, and that modification in the hydrophobic gate region by introducing a charged side chain between TM4/6 interface in TMEM16 and OSCA/TMEM63 family can allow lipid scramblase. In this manuscript, using scramblase activity via Annexin V binding to phosphatidylserine, and electrophysiology, the authors demonstrate that lysine mutation in TM4 of TMEM16F and TMEM16A can cause constitutive lipid scramblase activity. The authors then go on to show that analogous mutations in OSCA1.2 and TMEM63A can lead to scramblase activity. The revised version does a thorough characterization of residues that form the hydrophobic gate region in TM4/6 of this superfamily of channels. Their results indicated that disrupting the TM4/6 interaction can reduce energy barrier for this channels to scramblase lipids.

Strengths:

Overall, the authors introduce an interesting concept that this large superfamily can permeate ions and lipids.

Weaknesses:

none noted in the revised version.

---

## [Referee Report · Reviewer #2 (Public review)]

This focused study by Lowry and colleagues that identifies a key molecular motif that controls ion permeation vs combined ion permeation and lipid transport in three families of channel/scramblase proteins, in TMEM16 channels, in the plant-expressed and stress-gated cation channel OSCA, and in the mammalian homolog and mechanosensitive cation channel, TMEM63. Between them, these three channels share low sequence similarity and have seemingly differing functions, as anion (TMEM16 channels), or stress-activated cation channels (OSCA/TMEM63). The study finds that in all three families, mutating a single hydrophobic residue in the ion permeation pathway of the channels confers lipid transport through the pores of the channels, indicating that TMEM16 and related OSCA and TMEM63 channels have a conserved potential for both ion and lipid permeation. The authors interpret the findings as revealing that these channel/scramblase proteins have a relatively low "energetic barrier for scramblase" activity. The experiments are done with a high level of rigor and the revised paper is very well written and addresses the previous concerns.

---

## [Referee Report · Reviewer #3 (Public review)]

This study was focused on the conserved mechanisms across the Transmembrane Channel/Scramblase superfamily, which includes members of the TMEM16, TMEM63/OSCA, and TMC families. In previous work, the authors have studied the role of the inner activation gate of these proteins. Here, the authors show that the introduction of mutations at the TM4-TM6 interface, which are close to the inactivation gate, can disrupt gating and confer scramblase activity to non-scramblases proteins.

Overall, the confocal imaging experiments, patch clamping experiments, and data analysis are performed well and in line with standard methods. The molecular dynamics simulation work is focused but adds supportive evidence to their findings. Although there could have been more extensive molecular analysis to bolster the authors' arguments on the role of the TM4-TM6 interface (e.g. evaluate effects of size/hydrophobicity, double mutants, cross-linking, more in-depth simulation data), there is adequate evidence to conclude that certain residues at this interface is critical to ion conduction and phospholipid scramblase activity. The data presented only adds incremental depth of knowledge for each individual channel, but together, they show this to be true for conserved TM4 residues across TMEM16F, TMEM16A, OSCA1.2, and TMEM63A proteins. This breadth of data is a major strength of this paper, and provides strong evidence for a coupled pathway for ion conduction and phospholipid transport, though the underlying biophysical mechanism is still speculative and remains to be elucidated.

---

## [Author Response]

The following is the authors’ response to the original reviews.

**Reviewer #1 (Recommendations For The Authors):**
Figures 1 and 2. How do the authors know that the lysine mutations are specific to constitutive activity and not because it is causing the channel to be now voltage sensitive?

As shown in the revised Figs. 1b, S2a, and 3b, TMEM16F I521K/M522K, TMEM16F I521E, and TMEM16A I546K/I547K spontaneously expose PS, respectively. Neither membrane depolarization nor calcium stimulation was introduced under these conditions and the cells were grown in calcium-free media after transfection to limit calcium-dependent activation. Our new experiments further demonstrate that TMEM16F T526K (Fig. 1b) and TMEM16A E551K (Fig. 3b), which are further away from the activation gate, exhibit either strongly attenuated or lack spontaneous lipid scrambling activity. According to these results, the gain-of-function mutants (TMEM16FI521K/M522K/I521E and TMEM16A I546K/I547K) are indeed constitutively active. This constitutive scramblase activity is not due to a gain of voltage sensitivity as ion channel activity is also minimal around the resting membrane potential of a HEK cell (Fig. 1d, e and Fig. 3d, e).

The authors see very large currents of 5 -10 nA in their electrophysiology experiments in Figures 2D and 3D. I understand that Figure 2D are whole-cell recordings but are the authors confident that the currents that they are recordings from the mutants are indeed specific to TMEM16A. More importantly, in Figure 3D they see 3-5nA currents in insideout patches, which is huge. They have no added divalent in their bath solution, which could lead to larger single-channel amplitudes, but 3-5nA seems excessive. Some control to demonstrate that these are indeed OSCA1.2 currents is important.

TMEM16A and TMEM16F are well-known for their high cell surface expression. Therefore, the current amplitude is usually huge even in excised inside-out or outside-out patches—please see our previous publications for details: (1) 10.1016/j.cell.2012.07.036, (2) 10.7554/eLife.02772, (3) 10.1038/s41467-019-11784-8, (4) 10.1038/s41467-019-09778-7, (5) 10.1016/j.celrep.2020.108570, (6) 10.1085/jgp.202012704, and (7) 10.1085/jgp.202313460.

HEK293 cells do not have endogenous TMEM16A (https://doi.org/10.1038/nature07313, 10.1016/j.cell.2008.09.003 , DOI: 10.1126/science.1163518). It therefore serves as a widely used cell line for studying TMEM16A biophysics. As overexpressing the WT control barely elicited any obvious current in 0 Ca2+ (Fig. 3d), there is no doubt that the large outward-rectifying current (hallmark of CaCC) in the revised Fig. 3d (previous Fig. 2D) was elicited from the mutant TMEM16A channels. The strong outward rectification also rules out the possibility of this being leak current.

Regarding Fig. 4d (previous Fig. 3D), OSCA1.2 has excellent surface expression as shown in Fig. 4b. OSCA1.2 also has much higher single channel conductance (121.8 ± 3.4 pS, 10.7554/eLife.41844) than TMEM16A (~3-8 pS) and TMEM16F (<1 pS). Therefore, recording nA OSCA1.2 current from excised patches is normal given larger OSCA1.2 current at depolarized voltages than the current recorded at hyperpolarized voltages (please see our explanation in the next response). As the reviewer pointed out, lack of divalent ions in our experimental conditions may also partially contribute to the large conductance. To further verify, we conducted mock transfection recordings (please see Author response image 1 below). WT- but not mock (GFP)transfected cells gave rise to large current, further supporting that the recorded current was indeed through OSCA1.2.

**Author response image 1. sa4fig1:** Representative inside-out currents for mock (GFP)- and OSCA1. 2 WT-transfected cells. OSCA1.2 is responsible for nA currents elicited by the pressure and voltage protocols shown.

Figure 3D and 5D. Most of the traces and current quantification is done at positive potentials and is outward current. Do the authors observe inward currents? It is difficult to judge by the figures since currents are so large. OSCA/TMEM63s are cationic channels and all published data on these channels have demonstrated robust inward currents at negative, physiologically relevant potentials. The lack of inward currents but only large outward currents suggests that these mutations could be doing something else to the channel.

Yes. We indeed observe inward current at negative holding potentials under pressure clamp (Author response image 2). However, mechanosensitive OSCA and TMEM63A channels are also voltage dependent. Their outward current is an order of magnitude larger at depolarized voltages (*e.g.*, Author response image 2, also 10.7554/eLife.41844, see Fig. 1H).

**Author response image 2. sa4fig2:** Voltage-dependent rectification of OSCA1.2 current. a. Representative OSCA1.2 trace (bottom) elicited by a voltage-ramp under -50 mmHg (top). b. The difference in inward and outward current amplitudes.

We found that quantifying the OSCA1.2 outward current has advantages over the inward current. Usually, using the gold standard pressure clamp protocol at negative holding voltages, peak inward current amplitude is quantified. However, OSCA inward current quickly inactivates (10.7554/eLife.41844, see Fig. 1C). This makes robust quantification and comparison with mutant channels difficult. Holding the membrane at a constant pressure and measuring OSCA1.2 G-V overcomes these issues associated with the classical inward current measurements. The large depolarization-driven outward current does not inactivate, and robust tail current (Response Fig. 1, 2) allows us to construct G-V relationships. We found quantifying mutants’ voltage dependence at constant pressure is more consistent than quantifying pressure dependence at constant voltage. These advantages make our new protocol preferable to the commonly used gold standard pressure clamp protocol for characterizing and comparing the gating mutations identified in this manuscript.

Figure 3 and 5. Why are mechanically activated currents being recorded at random pressure stimuli (-50 mmHg for OSCA) and (-80 mmHg for Tmem63a)? The gold standard in the field is to run an entire pressure response curve. Given that only outward currents are observed at membrane potentials +120mV and above at 0mmHg, this questions whether they are indeed constitutively active.

As we explained in the previous response, both voltage and membrane stretch activate OSCA/TMEM63A channels. We found measuring voltage dependence under constant pressure provided more consistent quantification than the gold standard pressure response protocol. This may be due to the variability of applied membrane tension under repeated stretches versus the more consistent applied voltage. Additionally, we chose -50 mmHg and -80 mmHg to reflect the reported differences in half-maximal pressures between OSCA1.2 and TMEM63A (*e.g.*, P50 ~55 mmHg for 1.2 and ~61 mmHg for 63A in 10.7554/eLife.41844 versus ~86 mmHg for 1.2 and -123 mmHg for 63A in 10.1016/j.neuron.2023.07.006).

We also used higher pressure in cell attached mode to increase TMEM63A current amplitudes, which are usually tiny. We have updated our method section (Lines 329334) to further clarify why we used these protocols.

Please note that in TMEM16 proteins, ions and lipids might not always co-transport.

This means that under certain conditions, only one type of substrate may go through. For instance, in WT TMEM16F, Ca2+ stimulation can easily trigger PS exposure at resting membrane potential. No ionic currents are elicited until strong depolarization is applied. Similarly, the TMEM16F GOF mutations spontaneously transport lipids, leading to loss of lipid asymmetry (Fig. 1b, c). However, in 0 Ca2+, these TMEM16F mutant channels still need strong depolarization for ion conduction (Fig. 1d, e). Although the detailed mechanism still needs to be further investigated, the OSCA1.2 and TMEM63A GOF mutations share similar features with TMEM16 proteins, exhibiting ion conduction under high pressures and depolarizing voltages, yet constitutively active scrambling.

Some clarity is needed for their choice of residues. I understand that a lot of this is also informed by the structures of these ion channels. According to the alignment shown in Supplementary Figure 1, they chose LA for OSCA1.2, which is in line with the IM (TMEM16F) and II(TMEM16A) residues but for Tmem63a they chose the hydrophobic gate residue W and S. Was the A476 tested? Also, OSCA1.2 already has a K in the hydrophobic gating residue region. How do the authors reconcile this with their model?

We appreciate this critical comment. We have included the characterization of TMEM63A A476K (Fig. 6, corresponding to M522 in 16F, I547 in 16A, and A439 in OSCA1.2). Interestingly, A476K transfected cells did not show obvious spontaneous PS exposure yet exhibited a modest shift in V50 comparable to W472K and S475K. These differences may reflect the high-tension activated nature of the TMEM63 proteins (10.1016/j.neuron.2023.07.006) as compared to OSCA1.2, where the corresponding mutation (A439K, Fig. 4b, c) showed very little spontaneous activity and required hypotonic stimulation to promote more robust PS exposure (Fig. 5).

Furthermore, as we showed in Figs. 1b-c and 3b-c, there is a lower limit (towards the Cterminus) of the TM 4 lysine mutation effect, which becomes insufficient to cause a constitutively open pore for spontaneous lipid scrambling. It is possible that TMEM63A A476K represents the lower limit of TM 4 mutations that can convert TMEM63A into a spontaneous lipid scramblase.

Regarding OSCA1.2 K435 and TMEM63A W472, these sites correspond to the hydrophobic gate residues on TM 4 in TMEM16F (F518, Fig. 1a) and TMEM16A (L543, Fig. 3a) so it is unsurprising to us that a lysine mutation at this site causes constitutive scramblase activity in TMEM63A (Fig. 6b, c). For OSCA1.2, it is more intriguing since this residue is already a lysine (K435). In Supplementary Fig. 5 our new experiments show that neutralizing K435 with leucine (K435L) in the background of L438K significantly attenuates spontaneous PS exposure from ~63% PS positive for L438K alone (two lysine residues) to ~31% for K435L/L438K (one lysine). One the other hand, the K435L mutation by itself is also insufficient to induce PS exposure. Therefore, the endogenous lysine at residue 435 has an additive effect on the spontaneous scramblase activity of L438K. We believe the explanation for this result lies in experiments conducted in model transmembrane helices, which have shown that stacking hydrophilic side chains within the membrane interior promotes trans-bilayer lipid flipping (see 10.1248/cpb.c22-00133).

These same studies also support our observation (10.1038/s41467-019-09778-7) that highly hydrophilic side chains (such as lysine or glutamic acid) accelerate trans-bilayer lipid flipping more effectively than hydrophobic side chains such as isoleucine or alanine (Author response image 3, see also 10.1021/acs.jpcb.8b00298).

**Author response image 3. sa4fig3:** Trans-bilayer lipid flipping rates (kflip) accelerate with increasing side chain hydropathy for a residue placed in the center of a model transmembrane helical peptide.

How do the authors know that osmotic shock is indeed activating OSCA1.2 and TMEM63A? If they can record from the channels then electrophysiology data that confirms activation of the channel in the presence of hypoosmotic shock will strengthen the osmolarity active scramblase activity demonstrated in Figure 4. So far, there is conclusive data showing that they are mechanically activated but conclusive electrophysiological data for OSCA/TMEM63 osmolarity activation is not described yet, including the reference (38) they indicate in line 132. Although osmotic shock can perturb mechanical properties of the membrane it can also activate volume-regulated anion channels, which are also present in HEK cells.

Thank you for raising this important question. While reference 38, (now reference 39) shows direct electrophysiological evidence of hypertonicity-induced current (e.g., Fig. 4 f, g, i, and j in 10.1038/nature13593), direct electrophysiological evidence that OSCA/TMEM63 can be activated by hypotonic stimulation is still missing. To address this question, we conducted whole-cell patch clamp experiments on mocktransfected and OSCA1.2 WT-transfected cells stimulated with 120 mOsm/kg hypotonic solution, comparable to the same conditions as hypotonic-induced scrambling shown in Fig. 5. As shown in Supplementary Fig. 6, our whole-cell recording detected a slowly evolving yet robust outward rectifying current in OSCA1.2-transfected cells, which was not observed in mock transfected cells.

To avoid the contamination from endogenous SWELL osmo-/volume-regulated chloride channels, our new experiment used 140 mM Na gluconate to replace NaCl in both the pipette and the bath solution. Because SWELL/VRAC channels are minimally permeable to gluconate anions (*e.g.*, 10.1007/BF00374290), we conclude that hypotonic stimulation can indeed activate OSCA1.2 albeit with perhaps lower efficiency compared to mechanical stimulation.

Minor commentsWhat is the timeline for the scramblase assay for all the experiments (except Figure 4)? How long is the AnnexinV incubated before imaging?

Thank you for pointing out this point where we have not provided sufficient detail. Cells were imaged in the scramblase assay (including in Fig. 4, now revised Fig. 5) in AnnexinV-containing buffer immediately and without a formal incubation period because AnnexinV binding to exposed PS proceeds rapidly. We have included additional detail in the methods section to eliminate any confusion (Lines 310-312).

In some places of the document, it says OSCA/TMEM63, and in other places, it is denoted as TMEM63/OSCA. The literature so far has always called the family OSCA/TMEM63- please stay consistent with the field.

Thank you for pointing this out, we have corrected these instances to be consistent with the field.

**Reviewer #2 (Recommendations For The Authors):**
(1) The authors' statement that the channel/scramblase family members have a relatively low "energetic barrier for scramblase" activity needs further support. While mutating the hydrophobic channel gate certainly could destabilize ion conduction to cause a GOF effect on channel activity, it is still not clear why scramblase activity, which is tantamount to altered permeation, happens in the mutant channels. Are permeation and channel gating (opening) coupled in these channels? If so, what is the basis for the coupling? Is scramblase activity only observed when the gating is destabilized or are they separable?

We appreciate these great questions. For the question about the ‘energetic barrier’ statement, please see our response to point (3) where we have carried out MD simulations of the OSCA1.2 WT and L438K mutant to provide insight into how the permeation pathway is altered by these mutations.

Regarding why TMEM16A can be converted into a scramblase, we use the extensively studied TMEM16 proteins as examples to improve our current understanding of OSCA/TMEM63 proteins. For further details please see our original paper (10.1038/s41467-019-09778-7) and our review (10.3389/fphys.2021.787773), which are summarized as follows:

(1) The “neck region”, consisting of the exofacial halves of TMs 3-6, form the poregate region for both ion and lipid permeation (Author response image 4B). In the closed state, the neck region is constricted and TMs 4 and 6 interact with each other, preventing substrate permeation. The hydrophobic inner activation gate that we identified (10.1038/s41467-019-09778-7) resides right underneath the inner mouth of the neck region, controlling both ion and lipid permeation scrambling.

(2) Based on our functional observations and the available scramblase structures of TMEM16 proteins in multiple conformations, we proposed a clamshell-like gating model to describe TMEM16 lipid scrambling (Author response image 4D). According to this model, Ca2+-induced conformational changes weaken the TM 4/6 interface. This promotes the separation of the two transmembrane segments, analogous to the opening of a clam shell, allowing a membrane-spanning groove to facilitate permeation of the lipid headgroup.

(3) For the CaCC, TMEM16A, Ca2+ binding dilates the pore. However, the binding energy likely cannot open the TM 4/6 interface at the neck region so, in the absence of groove formation, only Cl- ions but not lipids can permeate. (Pore dilation model, Author response image 4C).

(4) Introducing charged residues near the inner activation gate disrupts the neck region, potentially by weakening the hydrophobic interactions between TMs 4 and 6. This mutational effect results in constitutively active TMEM16F scramblases and enables spontaneous lipid permeation in the TMEM16A CaCC.

(5) In our revision, we tested additional mutations with different side chain properties (Supplementary Fig. 2), validating previous findings by us (10.1038/s41467-01909778-7) and others (10.1038/s41467-022-34497-x) that gate disruption increases with the side chain hydropathy of the mutation.

(6) We further extended lysine mutations to two helical turns below the inner activation gate on TM 4 and identified a lower limit for mutation-induced spontaneous scramblase activity in TMEM16F and TMEM16A (Figs. 1b, c and 3b, c, respectively). Together, all these points lend additional support to our proposed gating models for TMEM16 proteins, which we postulate may also relate to the OSCA/TMEM63 family based on the evidence provided in our manuscript.

**Author response image 4. sa4fig4:** Model of gating (and regulatory) mechanisms in the TMEM16 family. (B) Overall architecture and proposed modules. (C) Pore-dilation gating model for CaCCs. (D) Clamshell gating model for CaPLSases.

Regarding the relationship between ion and lipid permeation through TMEM16 scramblases, the following is the summary of our current understanding:

(1) Functionally, ion and lipid permeation are not necessarily obligatory to each other. This is evidenced by our previous biophysical characterizations of TMEM16F ion channel and lipid scramblase activities. Ca2+ can trigger TMEM16F lipid scrambling at resting membrane potentials, however, Ca2+ alone is insufficient to record TMEM16F current. Strong membrane depolarization synergistically with elevated intracellular Ca2+ is required to activate ion permeation. Based on these observations, we postulate that ions and lipids may have different extracellular gates, despite sharing an inner activation gate (10.1038/s41467-019-09778-7). Ca2+ alone may sufficiently open the inner gate (and extracellular gate) for lipids, whereas depolarization is likely required to open the extracellular gate and allow ion flux. Further structure-function studies are needed to test this hypothesis.

(2) Structurally, the open conformation of TMEM16 scramblases such as the fungal orthologs and human TMEM16K (Supplementary Fig. 1 b-d) are widely open, which allows lipid and ion co-transport. Ion and lipid co-transport has also been demonstrated in various MD simulations (e.g., 10.7554/eLife.28671, 10.3389/fmolb.2022.903972, and 10.1038/s41467-021-22724-w)

(3) Functionally, we (10.1085/jgp.202012704) and others (10.7554/eLife.06901.001) have measured dual recording of channel and scramblase activities, also demonstrating that ions and lipids are co-transported simultaneously when the proteins are fully activated.

(4) In this manuscript, we also provide multiple examples (TMEM16F in Fig. 1, TMEM16A in Fig. 3, OSCA1.2 in Fig. 4, and TMEM63A in Fig. 6) of mutations showing spontaneous phospholipid scramblase activities, yet their channel activities require strong depolarization or, in the case of TMEM63A, high pressures to be elicited.

Together, this new evidence further supports our hypothesis that there might be multiple gates for ion and lipid permeation, in addition to the shared inner gate we previously identified. We hope these detailed explanations help convey the intricacy of these intriguing questions. Of course, future studies are needed to test our hypothesis and elucidate the complex relationship between ion and lipid permeation of these proteins.

(2) One weakness in the experimental approach is the very limited number of substitutions used to infer the conclusion regarding the energetic barrier and other conclusions relating to scramblase activity. Additional substitutions of charged and polar amino acids at the hydrophobic gate would be helpful in illuminating the molecular determinants of the GOF phenotype and also reveal varying patterns of lipid permeation which could be enormously informative. These additional mutations for analysis of TMEM16F and OSCA should be added to the study.

We appreciate these great suggestions which were shared by multiple reviewers. We have included our duplicated response below.

“Response to reviewers 2 & 3: In our 2019 paper (10.1038/s41467-019-09778-7), we have systematically tested the side chain properties at the inner activation gate of TMEM16F on lipid scrambling activity (Response Fig. 6) and, since then, these results have been supplemented by others as well (10.1038/s41467-022-34497-x). In summary, mutating the inner activation gate residues to polar or charged residues generally results in constitutively activated scramblases without requiring Ca2+ (Fig 5a in 10.1038/s41467-019-09778-7). Because these residues form a hydrophobic gate, introducing smaller side chains via alanine substitution are also gain-of-function with the Y563A mutant as well as the F518A/Y563A/I612A variant being constitutively active (Fig. 3a in 10.1038/s41467-019-09778-7). Meanwhile, mutating these gate residues to hydrophobic amino acids causes no change for I612W, a slight gain-of-function for F518W, slight loss-of-function of F518L, and complete loss-of-function for Y563W (Fig. 4b in 10.1038/s41467-01909778-7). These findings clearly demonstrate that the side-chain properties are critical for regulating the gate opening. Charged mutations including lysine and glutamic acid are the most effective to promote gate opening (Fig 5a in 10.1038/s41467-019-09778-7).

Similarly, others have observed that side chain hydropathy at the F518 site in TMEM16F correlates with shifts in the Ca2+ EC50 (Fig. 2 of 10.1038/s41467-022-34497-x). Note that this publication resolved the structure of the TMEM16F F518H mutant, revealing a previously unseen conformation that we have highlighted in Supplementary Fig. 1e and discussed in lines 235-238. Please also see our response to Reviewer #1 above, where we discuss discoveries in model transmembrane helical peptide systems showing that transbilayer lipid flipping rates correlate with side chain hydropathy (Author response image 3), distance between stacked hydropathic residues (schematic in 10.1248/cpb.c22-00133), and even helical angle between stacked side chains (not show).

Following the reviewers’ suggestions, we have tested additional mutations in alternative locations and with different side chains.

(1) We have added data for TMEM16F I521A and I521E to demonstrate a similar effect of alternative side chains to what has previously been reported by us and others. We found that I521A failed to show spontaneous scrambling activity (Supplementary Fig. 2), yet I521E (Supplementary Fig. 2) is a constitutively active lipid scramblase, similar to I521K (Fig. 1). This further demonstrates that gate disruption correlates with the side chain hydropathy and that this site lines a critical gating interface.

(2) We also added lysine mutations two helical turns below the conserved inner activation gate for TMEM16F T526 (Fig. 1), TMEM16A E551 (Fig. 3). We found that there is indeed a lower limit for the observed effect in TMEM16, where lysine mutations no longer induce spontaneous lipid scrambling activity. This indicates that when TM 4/6 interaction is weaker toward intracellular side (Figs. 1a, 3a), the TM 4 lysine mutation loses the ability to promoting lipid scrambling by disrupting the TM 4/6 interface to enable clamshell-like opening of the permeation pathway.

(3) We added a TMEM16F lysine mutation on TM 6 at residue I611 (Fig. 2). Similar to I612K (Response Fig. 6), I611K also leads to spontaneous lipid scrambling and enhanced channel activity in the absence of calcium (Fig. 2). This shows that charged mutations along TM 6 can also promote lipid scrambling, strengthening our model that hydrophobic interactions along the TM 4/6 interface are critical for gating and lipid permeation.”

(3) Related to the above point, it would be enormously useful to perform even limited computational modelling to support the "energetic barrier" statement. Specifically, can the authors model waters in the putative pore to examine water occupancy in the WT and mutant channels to better understand how the barrier for ions and lipids is altered in the TMEM16?

We appreciate this suggestion and have now conducted atomistic MD simulations of OSCA1.2 WT and L438K mutant for ~1 μs (Supplementary Fig. 4). The simulations revealed, elevated water occupancy in the pore region of the L438K mutant, likely due to a widening at the TM 4/6 interface. Conversely, the WT interface remained constricted, largely disallowing water occupancy. These computational results support our previously proposed clamshell-like gating model for TMEM16 scramblases and provide strong support that the L438K mutation is disrupting the interaction of the TM 4/6 interface, in turn reducing the energetic barrier for both ion and lipid permeation.

(4) I am puzzled about the ability of OSCA and the TMEM63 proteins which are cation channels to conduct negatively charged lipids. How can the pore be selective for cations and yet permeate negatively charged molecules when lipids are presented?

This is a great question. TMEM16 scramblase (as well as other known scramblases, such as the Xkr and Opsin families) are surprisingly non-selective to phospholipids (all major phospholipid species, not just anionic lipids like PS). It is still debated whether lipid headgroups indeed insert into an open pore or hydrophilic groove (Response Fig. 5), or if they may traverse the bilayer by the so-called ‘out-of-groove’ model. Regardless of the model, the consensus is that Ca2+-induced conformational changes catalyze lipid permeation and the mutations we have introduced are designed to mimic these conformational changes by separating the TM 4/6 interface.

Additionally, TMEM16F channel activity was first characterized as cation non-selective (10.1016/j.cell.2012.07.036), similar to OSCA/TMEM63s, which may even exhibit some chloride permeability (10.7554/eLife.41844.001). Thus, it appears as though scramblase activity is agnostic to headgroup charge and compatible with both a mutant anion channel (TMEM16A) and mutant cation channels (TMEM16F, OSCA1.2, and TMEM63A), however, more detailed structural, functional, and computational studies are needed to further clarify ion and lipid co-transport mechanisms.

(5) Do pore blockers like Gd3+ which block permeation also inhibit the scramblase activity of the mutant channels? This should be tested for the mutant channels.

While extracellular Gd3+ has been previously reported as an inhibitor of OSCA1.2 (10.7554/eLife.41844.001), we did not observe this effect (Author response image 5), but instead saw inhibition by intracellular Gd3+ (Author response image 6). Given this discrepancy, we did not test Gd3+ inhibition of the OSCA1.2 scramblases, but instead tested Ani9, a paralog-specific inhibitor of TMEM16A, on the TMEM16A I546K gain-offunction and found it attenuated both ion channel and phospholipid scramblase activities (Supplementary Fig. 3).

**Author response image 5. sa4fig5:** 200 µM Gd3+ext fails to inhibit OSCA1.2 currents in cell-attached patches. Pressure-elicited peak currents (n=6 each). Statistical test is an unpaired Student’s t-test.

**Author response image 6. sa4fig6:** 200 µM Gd3+int completely inhibits OSCA1.2 currents in inside-out patches. (a) Representative traces in before (black), during (red), and after (blue) Gd3+ application. (b) Representative application timecourse. (c) Quantification of peak currents (n=8 each). Statistical test is one-way ANOVA.

Minor:- Some of the current amplitudes shown in Figures 2 and 3 are enormous. Is liquid junction potential corrected in these experiments? If not, it would be preferable to correct this to avoid voltage errors.

Thanks for the question. The large current amplitude is due to (1) great surface expression of the proteins; (2) large single channel conductance of OSCA channels, (3) much larger current at positive voltages for OSCA channels. Our control experiment showed that WT TMEM16A at 0 Ca2+ did not give rise to any current (Fig. 3d), further demonstrating that the large current was not due to liquid junction potential. For the OSCA recordings, we also did not observe current in mock-transfected cells, further excluding the possible interference of liquid junction potential (Response Fig. 1)

- Related, authors could consider adding some evidence using selective pharmacology to support the conclusions that the observed currents arise from TMEM or OSCA channels.

Thanks for the suggestion. As mentioned above, we have added experiments with Ani9, a specific inhibitor of TMEM16A, in Supplementary Fig. 3. We found that Ani9 robustly attenuated both ion channel and phospholipid scramblase activities for the TMEM16A I546K gain-of-function mutant. This is also consistent with our previous publication (10.1038/s41467-019-09778-7), where Ani9 efficiently inhibited the TMEM16A L534K mutant scramblases. Additionally, we have provided mock controls (Response Fig. 1, Fig. 6d, e) to show that the observed currents are indeed attributable to OSCA1.2 and TMEM63A.

**Reviewer #3 (Recommendations For The Authors):**
Given that the authors postulate that the introduction of a positive charge via the lysine side chain is essential to the constitutive activity of these proteins, additional mutation controls for side chain size (e.g. glutamine/methionine) or negative charge (e.g. glutamic acid), or a different positive charge (i.e. arginine) would have strengthened their argument. To more comprehensively understand the TM4/TM6 interface, mutations at locations one turn above and one turn below could be studied until there is no phenotype. In addition, the equivalent mutations on the TM6 side should be explored to rule out the effects of conformational changes that arise from mutating TM4 and to increase the strength of evidence for the importance of side-chain interactions at the TM6 interface.

We appreciate these great suggestions which were shared by multiple reviewers. We have included our previous responses below.

“Response to reviewers 2 & 3: In our 2019 paper (10.1038/s41467-019-09778-7), we have systematically tested the side chain properties at the inner activation gate of TMEM16F on lipid scrambling activity (Response Fig. 6) and, since then, these results have been supplemented by others as well (10.1038/s41467-022-34497-x). In summary, mutating the inner activation gate residues to polar or charged residues generally results in constitutively activated scramblases without requiring Ca2+ (Fig 5a in 10.1038/s41467-019-09778-7). Because these residues form a hydrophobic gate, introducing smaller side chains via alanine substitution are also gain-of-function with the Y563A mutant as well as the F518A/Y563A/I612A variant being constitutively active (Fig. 3a in 10.1038/s41467-019-09778-7). Meanwhile, mutating these gate residues to hydrophobic amino acids causes no change for I612W, a slight gain-of-function for F518W, slight loss-of-function of F518L, and complete loss-of-function for Y563W (Fig. 4b in 10.1038/s41467-01909778-7). These findings clearly demonstrate that the side-chain properties are critical for regulating the gate opening. Charged mutations including lysine and glutamic acid are the most effective to promote gate opening (Fig 5a in 10.1038/s41467-019-09778-7).

Similarly, others have observed that side chain hydropathy at the F518 site in TMEM16F correlates with shifts in the Ca2+ EC50 (Fig. 2 of 10.1038/s41467-022-34497-x). Note that this publication resolved the structure of the TMEM16F F518H mutant, revealing a previously unseen conformation that we have highlighted in Supplementary Fig. 1e and discussed in lines 235-238. Please also see our response to Reviewer #1 above, where we discuss discoveries in model transmembrane helical peptide systems showing that transbilayer lipid flipping rates correlate with side chain hydropathy (Author response image 3), distance between stacked hydropathic residues (schematic in 10.1248/cpb.c22-00133), and even helical angle between stacked side chains (not show).

Following the reviewers’ suggestions, we have tested additional mutations in alternative locations and with different side chains.

(1) We have added data for TMEM16F I521A and I521E to demonstrate a similar effect of alternative side chains to what has previously been reported by us and others. We found that I521A failed to show spontaneous scrambling activity (Supplementary Fig. 2), yet I521E (Supplementary Fig. 2) is a constitutively active lipid scramblase, similar to I521K (Fig. 1). This further demonstrates that gate disruption correlates with the side chain hydropathy and that this site lines a critical gating interface.

(2) We also added lysine mutations two helical turns below the conserved inner activation gate for TMEM16F T526 (Fig. 1), TMEM16A E551 (Fig. 3). We found that there is indeed a lower limit for the observed effect in TMEM16, where lysine mutations no longer induce spontaneous lipid scrambling activity. This indicates that when TM 4/6 interaction is weaker toward intracellular side (Figs. 1a, 3a), the TM 4 lysine mutation loses the ability to promoting lipid scrambling by disrupting the TM 4/6 interface to enable clamshell-like opening of the permeation pathway.

(3) We added a TMEM16F lysine mutation on TM 6 at residue I611 (Fig. 2). Similar to I612K (Response Fig. 6), I611K also leads to spontaneous lipid scrambling and enhanced channel activity in the absence of calcium (Fig. 2). This shows that charged mutations along TM 6 can also promote lipid scrambling, strengthening our model that hydrophobic interactions along the TM 4/6 interface are critical for gating and lipid permeation.”

The experiments for OSCA1.2 osmolarity effects on gating and scramblase in Figure 4 could be improved by adding different levels of osmolarity in addition to time in the hypotonic solution.

We thank the reviewer for this excellent suggestion. We extensively tested this idea and found evidence (Response Fig. 10) that intermediate osmolarity (220 and 180 mOso/kg) also can enhance the scramblase activity of the A439K mutant, albeit to a milder extent compared to 120 mOso/kg stimulation. This suggests that swellinginduced membrane stretch may proportionally induce A439K activation and lipid scrambling. Due to the relatively mild sensitivity of OSCA to osmolarity and the variations induced by the experimental conditions, we believe it is better to not include this data to avoid overclaiming. We hope the reviewer would agree.

**Author response image 7. sa4fig7:** AnV intensities of WT- and A439K-transfected cells after 10 minutes of hypotonic stimulation at the listed osmolarities.

Some confocal images appear to be rotated relative to each other (e.g. Figures 2b and 3b).

**T**hank you for identifying these errors, they are corrected in the revision.